# Generate Any Scene: Synthetic Training and Evaluation Data for Generating Visual Content

**Ziqi Gao**[1],[*] **Weikai Huang**[1],[*] **Jieyu Zhang**[1]**, Aniruddha Kembhavi**[2]**, Ranjay Krishna**[1],[2]

[1]University of Washington, [2]Allen Institute of Artificial Intelligence

◯ Code: `https://github.com/RAIVNLab/GenerateAnyScene`

🤗 Dataset: GenerateAnyScene Dataset

## Abstract

Recent advances in text-to-vision generation excel in visual fidelity but struggle with compositional generalization and semantic alignment. Existing datasets are noisy and weakly compositional, limiting models' understanding of complex scenes, while scalable solutions for dense, high-quality annotations remain a challenge. We introduce GENERATE ANY SCENE, a data engine that systematically enumerates scene graphs representing the combinatorial array of possible visual scenes. GENERATE ANY SCENE dynamically constructs scene graphs of varying complexity from a structured taxonomy of objects, attributes, and relations. Given a sampled scene graph, GENERATE ANY SCENE translates it into a caption for text-to-image or text-to-video generation; it also translates it into a set of visual question answers that allow automatic evaluation and reward modeling of semantic alignment. Using GENERATE ANY SCENE, we first design a self-improving framework where models iteratively enhance their performance using generated data. *SDv1.5* achieves an average **4%** improvement over baselines and surpassing fine-tuning on CC3M. Second, we also design a distillation algorithm to transfer specific strengths from proprietary models to their open-source counterparts. Using fewer than 800 synthetic captions, we fine-tune *SDv1.5* and achieve a **10%** increase in TIFA score on compositional and hard concept generation. Third, we create a reward model to align model generation with semantic accuracy at a low cost. Using GRPO algorithm, we fine-tune SimpleAR-0.5B-SFT and surpass CLIP-based methods by **+5%** on DPG-Bench. Finally, we apply these ideas to the downstream task of content moderation where we train models to identify challenging cases by learning from synthetic data.

## 1 Introduction

Despite the high-fidelity of modern generative models (text-to-image and text-to-video), we are yet to witness wide-spread adoption [1, 2, 3, 4, 5]. Controllability remains out of reach [6]. Generated content appears realistic but often falls short of semantic alignment [7, 8, 9, 10]. Users prompt models with a specific concept in mind. For example, when prompted to generate a scene of a "A black dog chasing after a rabbit that is eating the grass, in Van Gogh's style, with starlight lightening", some models are likely to generate an image of a dog but might miss the rabbit or get the style incorrect.

We hypothesize that these limitations stem not only from architectural bottlenecks but more fundamentally from the lack of structured, compositionally rich training data [3], especially those with uncommon compositions. Popular datasets such as LAION [11] and CC3M [12] predominantly consist of web-crawled image-caption pairs, which are inherently noisy, weakly compositional, and biased toward single-object, coarse-grained descriptions. Such datasets lack explicit grounding of object-attribute relations and multi-object interactions, restricting models' ability to generalize to

complex visual scenes. Efforts to enhance caption quality [3, 13] have demonstrated that enhancing the compositional density and semantic richness of captions can significantly improve generative performance. Nevertheless, manual curation of such dense compositional annotations is labor-intensive, while automatic annotation methods (e.g., via MLMs) suffer from hallucination and semantic noise.

Constructing a compositional dataset requires that we first define *the space of the visual content*. Scene graphs are one such representation of the visual space [14, 15, 16, 17, 18], grounded in cognitive science [19]. A scene graph represents objects in a scene as individual nodes in a graph. Each object is modified by attributes, which describe its properties. For example, attributes can describe the material, color, size, and location of the object in the scene. Finally, relationships are edges that connect the nodes. They define the spatial, functional, social, and interactions between objects [20]. For example, in a living room scene, a "table" node might have attributes like "wooden" or "rectangular" and be connected to a "lamp" node through a relation: "on top of". This systematic scene graph structure provides simple yet effective ways to define and model the scene. As such, scene graphs are an ideal foundation for systematically defining the compositional space of visual content in text-to-vision generation.

We introduce GENERATE ANY SCENE, a system capable of efficiently enumerating the space of scene graphs representing a wide range of visual scenes. GENERATE ANY SCENE composes scene graphs of any structure using a rich taxonomy of visual elements, translating each scene graph into an input caption and visual question answers to evaluate the output image or video. In particular, we first construct a rich taxonomy of visual concepts consisting of $28,787$ objects, $1,494$ attributes, $10,492$ relations, $2,193$ scene attributes from various sources. Based on these assets, GENERATE ANY SCENE can synthesize an almost infinite number of scene graphs of varying complexity [21]. Besides, GENERATE ANY SCENE allows configurable scene graph generation. For example, evaluators can specify the complexity level of the scene graph to be generated or provide a seed scene graph to be expanded. By automating these steps, our system ensures both scalability and adaptability, providing researchers and developers with diverse, richly detailed scene graphs and corresponding captions tailored to their specific needs. We also conduct comprehensive text-to-vision evaluations using our generated captions, as detailed in Appendix A.

We show that GENERATE ANY SCENE can allow generation models to self-improve. Our diverse captions can facilitate a framework to iteratively improve *Text-to-Vision generation* models using their own generations. Given a model, we generate multiple images, identify the highest-scoring one, and use it as new fine-tuning data to improve the model itself. We fine-tune *SDv1.5* [22] and achieve an average of **4%** performance boost compared with original models, and this method is even better than fine-tuning with the same amount of real images and captions from the Conceptual Captions CC3M over different benchmarks.

We also use GENERATE ANY SCENE to design targeted distillation algorithms. Using our evaluations, we identify limitations in open-sourced models that their proprietary counterparts excel at. Next, we distill these specific capabilities from proprietary models. For example, *DaLL-E 3* [3] excels particularly in generating composite images with multiple parts. We distill this capability into *SDv1.5*, effectively bridging the gap between *DaLL-E 3* and *SDv1.5*. After targeted fine-tuning, *SDv1.5* achieves a **10%** increase in TIFA score [23] for compositional tasks and hard concept generation.

Then we propose a low-cost scene graph-based reward model for RLHF [24] in text-to-image generation. By leveraging synthetic scene graphs generated by GENERATE ANY SCENE, we generate exhaustive question-answer pairs that cover all objects, attributes, and relationships in the caption. Our method enables fine-grained, compositional reward modeling without manual annotation or heavy LLM inference. With GRPO [25], we fine-tune SimpleAR-0.5B-SFT [26] using a scene graph reward model, achieving better compositional alignment than CLIP-based methods [27] (**+5%** on DPG-Bench [28]).

Finally, we apply GENERATE ANY SCENE to the downstream application of content moderation. Content moderation is a vital application, especially as *Text-to-Vision generation* models improve. A key challenge lies in the limited diversity of existing training data. To address this, we leverage GENERATE ANY SCENE to generate diverse and compositional captions, creating synthetic training data that complements existing datasets. By retraining a ViT-T [29] detector with our enriched dataset, we enhance its detection performance, particularly in cross-model and cross-dataset scenarios.

## 2 Generate Any Scene

In this section, we present GENERATE ANY SCENE (Figure 1), a data engine that systematically synthesizes diverse scene graphs in terms of both structure and content and translates them into corresponding captions.

**Scene graph.** A scene graph is a structured representation of a visual scene, where objects are represented as nodes, their attributes (such as color and shape) are properties of those nodes, and the relationships between objects (such as spatial or semantic connections) are represented as edges. In recent years, scene graphs have played a crucial role in visual understanding tasks, such as those found in Visual Genome [14] and GQA [30] for visual question answering (VQA). Their utility has expanded to various *Text-to-Vision generation* tasks. For example, the DSG [31] and DPG [10] benchmarks leverage scene graphs to evaluate how well generated images align with captions.

**Taxonomy of visual elements.** To construct a scene graph, we use three main metadata types: **objects**, **attributes**, and **relations**. We further introduce **scene attributes** that capture global visual contexts, such as art style, to facilitate comprehensive caption synthesis. The statistics and source of our metadata are shown in Table 1. Additionally, we build a hierarchical taxonomy that categorizes metadata into distinct levels and types, enabling fine-grained analysis. This structure supports precise content synthesis, from broad concepts like "flower" to fine-grained instances such as "daisy."

Table 1: Summary of the quantities and sources of visual elements.

| Metadata Type | Number | Source |
| --- | --- | --- |
| Objects | 28,787 | WordNet [32] |
| Attributes | 1,494 | Wikipedia [33], etc. |
| Relations | 10,492 | Synthetic Visual Genome [34] |
| Scene Attributes | 2,193 | Places365 [35], etc. |

### 2.1 Generating data with scene graphs

**Step 1: Scene graph structure enumeration and query.** Our engine first generates and stores a variety of scene graph structures based on a specified level of **structural constraints**, such as complexity [36], average degree and the number of connected components. defined by the total number of objects, relationships, and attributes in each graph. The process begins by determining the number of object nodes, and then by systematically enumerating different combinations of relationships among these objects and their associated attributes. Once all graph structures satisfying the given constraints are enumerated, they are stored in a database for later use. This enumeration process is executed only once for each combination of structural parameters, allowing us to efficiently query the database for suitable templates when needed.

**Step 2: Populate the scene graph structure with metadata.** Given a generated scene graph structure, the next step involves populating the graph with metadata. For each object node, attribute node, and relation edge, we sample the corresponding content from our metadata. This process is highly customizable and controllable: users can define the topics and types of metadata to include, for instance, by selecting only commonsense metadata or specifying relationships between particular objects. By determining the scope of metadata sampling, we can precisely control the final content of the captions and easily extend the diversity and richness of scene graphs by adding new metadata.

**Step 3: Sample scene attributes.** We also include scene attributes that describe aspects such as the art style, viewpoint, time span (for video), and 3D attributes (for 3D content). These scene attributes are sampled directly from our metadata, creating a list that provides contextual details to enrich the description of the visual content.

**Step 4: Translate scene graph to caption.** We introduce an algorithm that converts scene graphs and a list of scene attributes into captions. The algorithm processes the scene graph in topological order, transforming each object, its attributes, and relational edges into descriptive text. To maintain coherence, it tracks each concept's occurrence, distinguishing objects with identical names using terms like "the first" or "the second." Objects that have been previously referenced without new

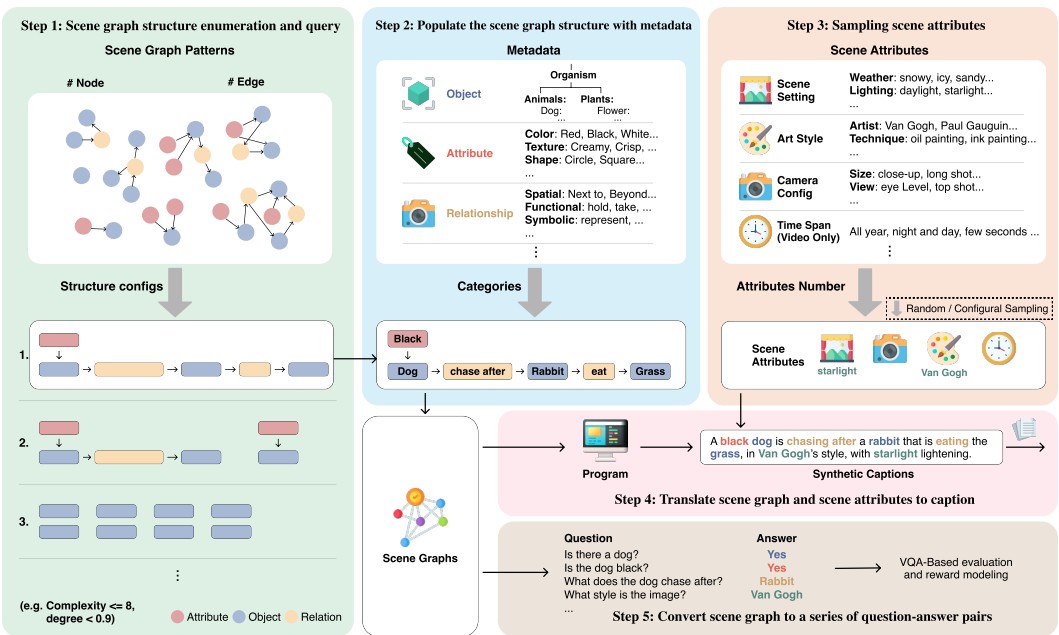

Figure 1: The generation pipeline of GENERATE ANY SCENE. **Step 1:** Enumerate diverse scene graph structures under user-defined constraints. **Step 2:** Populate structures with sampled objects, attributes, and relations. **Step 3:** Sample scene attributes such as style, perspective, or time span. **Step 4:** Translate scene graph and attributes into coherent captions. **Step 5:** Automatically generate QA pairs covering all elements for evaluation and reward modeling.

relations are skipped to avoid misreferencing. This approach enhances caption clarity by preventing repetition and maintaining a logical reference.

**Step 5: Convert scene graph to a series of question-answer pairs.** Given a synthetic scene graph, GENERATE ANY SCENE supports systematically enumerating exhaustive question-answer (QA) pairs that cover every compositional element. For instance, GENERATE ANY SCENE can generate questions about object attributes (e.g., What color is the sphere?), spatial relationships (e.g., What is to the left of the cube?), and so on, where each answer corresponds to a node (object or attribute) or an edge (relationship) in the scene graph. This method ensures comprehensive coverage of all objects, attributes, and relationships described in the caption, with negligible computational overhead. By automating this process, one can not only leverage VQA-based metrics [37, 31] to evaluate the generated images, but also construct a fine-grained, compositional reward model without requiring manual annotations or costly LLM inference.

# 3    Self-Improving models with synthetic captions

With GENERATE ANY SCENE, we develop a self-improvement framework to improve generative capabilities. By generating scalable compositional captions from scene graphs, GENERATE ANY SCENE expands the textual and visual space, allowing for a diversity of synthetic images that extend beyond real-world scenes. Our goal is to utilize these richly varied synthetic images to further boost model performance.

**Iterative self-improving framework.** Inspired by DreamSync [39], we designed an iterative self-improving framework using GENERATE ANY SCENE with *SDv1.5* as the baseline model. With *VQA Score*, which shows strong correlation with human evaluations on compositional images [37], we guide the model's improvement throughout the process. Specifically, GENERATE ANY SCENE generates $3 \times 10K$ captions across three epochs. For each caption, *SDv1.5* generates 8 images, and the image with the highest *VQA Score* is selected. From each set of 10K optimal images, we then select the top 25% (2.5K image-caption pairs) as the training data for each epoch. In subsequent

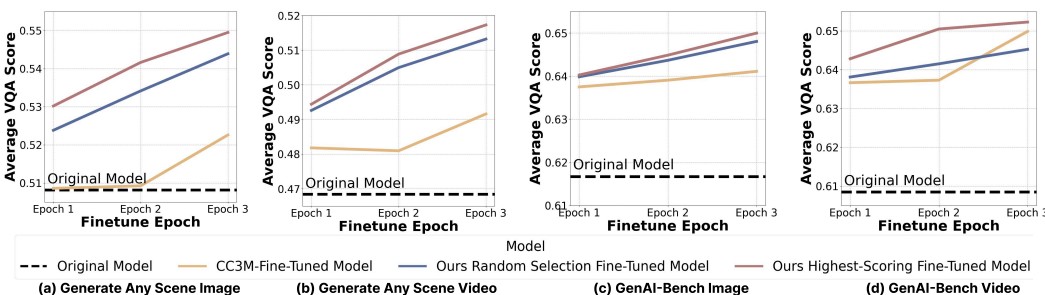

Figure 2: **Results for Self-Improving Models**. Average VQA score of *SDv1.5* fine-tuned on different data across 1K GENERATE ANY SCENE image/video evaluation set and GenAI-Bench image/video benchmark [38].

epochs, we use the fine-tuned model from the prior iteration to generate new images. We employ LoRA [40] for parameter-efficient fine-tuning.

**Baselines.** We conduct comparative experiments with the CC3M dataset, which comprises high-quality and diverse real-world image-caption pairs [12]. We randomly sample $3 \times 10K$ captions from CC3M, applying the same top-score selection strategy for iterative fine-tuning of *SDv1.5*. Additionally, we include a baseline using random-sample fine-tuning strategy to validate the advantage of our highest-scoring selection-based strategy. We evaluate our self-improving pipeline on *Text-to-Vision generation* benchmarks, including GenAI Bench [38]. For the *Text-to-Video generation* task, we use *Text2Video-Zero* as the baseline model, substituting its backbone with the original *SDv1.5* and our fine-tuned *SDv1.5* models.

**Fine-tuning with our synthetic captions can surpass high-quality real-world image-caption data.** Our results show that fine-tuning with GENERATE ANY SCENE-generated synthetic data consistently outperforms CC3M-based fine-tuning across *Text-to-Vision generation* tasks (Figure 2), achieving the highest gains with our highest-scoring selection strategy. This highlights GENERATE ANY SCENE's scalability and compositional diversity, enabling models to effectively capture complex scene structures. Additional experiment settings and results are in Appendix C.

# 4 Distilling targeted capabilities

Although self-improving with GENERATE ANY SCENE shows clear advantages over high-quality real-world datasets, its efficiency is inherently limited by the model's own generation capabilities. To address this, we leverage the taxonomy and systematical generation capabilities within GENERATE ANY SCENE to identify specific strengths of proprietary models (*DaLL-E 3*), and distill these capabilities into open-source models. More details are in Appendix D.

We evaluate multiple models using GENERATE ANY SCENE controllably generated captions and observe that *DaLL-E 3* achieves *TIFA Score* **1.5** to **2** times higher than those of other models. As shown in Figure 4a, when comparing *TIFA Score* across captions with varying numbers of elements (objects, relations, and attributes), *DaLL-E 3* **counterintuitively** maintains consistent performance regardless of element count. The performance of other models declines as the element count increases, which aligns with expected compositional challenges. We suspect that these differences are primarily due to *DaLL-E 3*'s advanced capabilities in compositionality and **understanding hard concepts**, which ensures high faithfulness across diverse combinations of element types and counts.

**Distilling compositionality from DaLL-E 3.** When analyzing model outputs from our synthetic captions, we find that *DaLL-E 3* tends to produce straightforward combinations of multiple objects (Figure 3). In contrast, open-source models like *SDv1.5* often omit objects from the captions, despite being capable of generating each one individually. This difference suggests that *DaLL-E 3* may benefit from training data emphasizing multi-object presence, even without detailed layout or object interaction. Such training likely underpins *DaLL-E 3*'s stronger performance on metrics like *TIFA Score* and *VQA Score* that prioritize object inclusion. To effectively distill these compositional

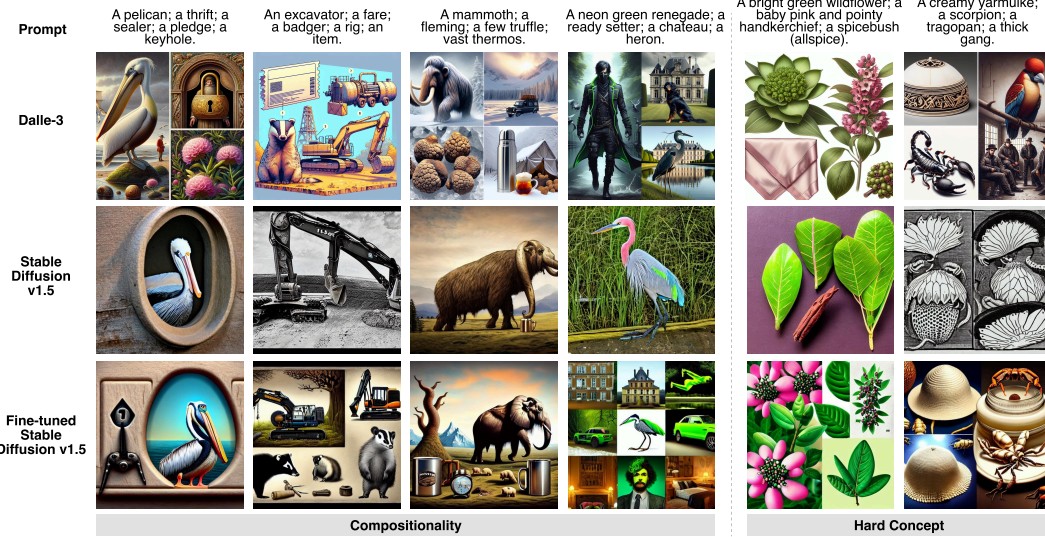

Figure 3: **Examples for Distilling Capabilities.** Examples of images generated by *DaLL-E 3*, the original *SDv1.5*, and the fine-tuned versions. The left four captions demonstrate fine-tuning with multi-object captions generated by GENERATE ANY SCENE for better compositionality, while the right two columns focus on understanding hard concepts.

abilities into *SDv1.5*, we employ GENERATE ANY SCENE for targeted synthesis of 778 multi-object captions, paired with images generated by *DaLL-E 3*, for finetuning *SDv1.5*.

**Distilling hard concepts understanding from DaLL-E 3.** Figure 3 shows that *DaLL-E 3* is capable not only of handling multi-object generation but also of understanding and generating rare and hard concepts, such as a specific species of flower. We attribute this to its training with proprietary real-world data. Using the taxonomy of GENERATE ANY SCENE, we compute model performance on each concept by averaging generation scores across captions containing that concept. Accumulating results through the taxonomy, we identify the 100 concepts where *SDv1.5* shows the largest performance gap relative to *DaLL-E 3*. For distilling, we generate 778 captions incorporating these hard concepts with other elements, and use *DaLL-E 3* to produce corresponding images.

**Baselines.** For the baseline, we randomly synthesize 778 captions using GENERATE ANY SCENE paired with *DaLL-E 3*-generated images to fine-tune the model. To evaluate model improvements, we generate another 1K multi-object captions and 1K hard-concept captions separately.

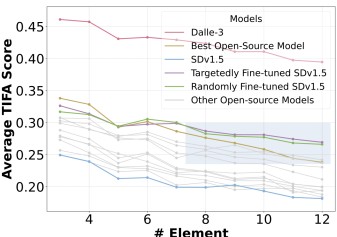 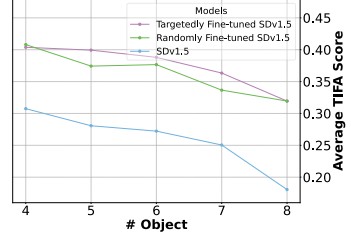 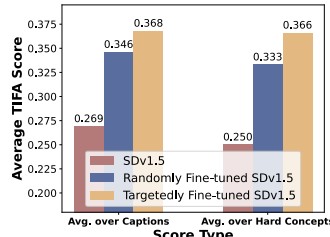

(a) **Distilling compositionality from DaLL-E 3**: Model results on TIFA vs. total element numbers in captions in 10K general GENERATE ANY SCENE captions.

(b) **Distilling compositionality from DaLL-E 3**: Model results on TIFA vs. total element numbers in captions in 1K multi-object GENERATE ANY SCENE captions.

(c) **Distilling hard concepts understanding from DALL-E 3**: Models' average *TIFA Score* performance over captions and hard concepts in 1K hard concepts GENERATE ANY SCENE captions.

Figure 4: **Results for Distilling Capabilities**. The left two figures show the results for **Distilling compositionality**, while the rightmost figure shows the results for **Distilling hard concepts understanding from DALL-E 3**.

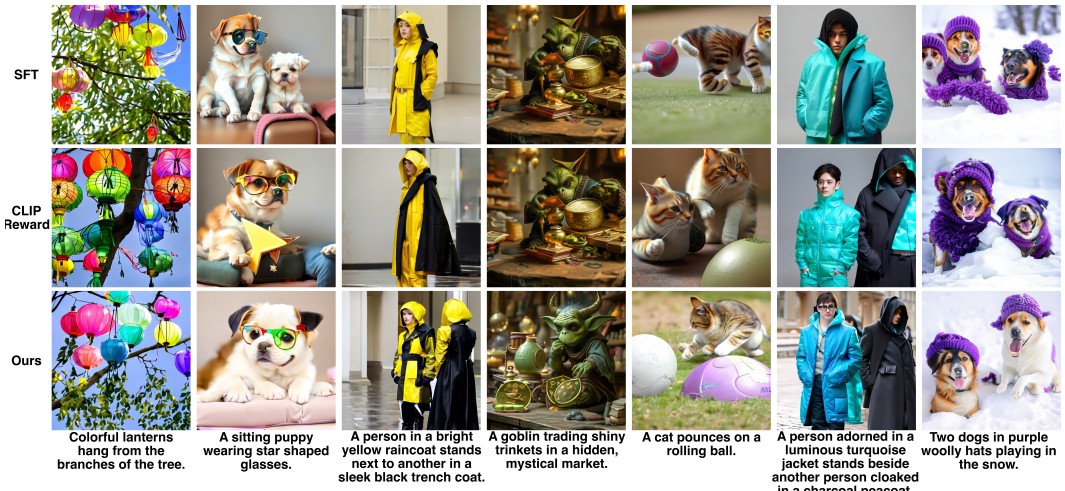

Figure 5: **Comparison of generated images.** Our reward model enables image generation with better semantic alignment, realism, and visual quality than baselines.

**Targeted caption synthesis via GENERATE ANY SCENE enables effective distillation of compositional abilities and hard concept understanding.** We analyze images generated by *SDv1.5* before and after fine-tuning on high-complexity captions (Figure 3). Surprisingly, with fewer than 1K LoRA fine-tuning steps, *SDv1.5* effectively learns *DaLL-E 3* 's capability to arrange and compose multiple objects within a single image. Quantitatively, Figure 4b shows a 10% improvement in *TIFA Score* after targeted fine-tuning, surpassing the performance of the randomly fine-tuned model. On a broader set of 10K GENERATE ANY SCENE-generated captions, the targeted fine-tuned model consistently outperforms randomly fine-tuned and original counterparts across complex scenes (Figure 4a). These results confirm not only the effectiveness but also the scalability and efficiency of GENERATE ANY SCENE. Also, the results in Figure 4c show that our targeted fine-tuning with hard concepts leads to improved model performance, reflected in higher average scores across captions and increased scores for each challenging concept.

## 5   Reinforcement learning with a synthetic reward function

Reinforcement Learning with Human Feedback (RLHF) has become an increasingly popular fine-tuning strategy in text-to-image generation [41, 42, 26]. However, defining an effective reward model that accurately captures semantic alignment for text-to-image generation remains an open challenge. Existing reward models like CLIP offer only coarse-grained image-text similarity signals, which fall short in assessing compositional correctness and lack interpretability. Alternative approaches have explored using visual question answering (VQA) as a proxy for evaluating semantic alignment, aiming for finer-grained assessments, yet require either labor-intensive datasets with dense annotations or large volumes of contextually relevant questions via advanced LLMs. Leveraging its structured scene graph synthesis capabilities, GENERATE ANY SCENE offers a scalable alternative by producing exhaustive semantic queries with negligible overhead, enabling low-cost, compositional reward modeling (Sec 2.1).

**Experiment setup.** Building on this scene graph-based reward modeling strategy, we adopt Group Relative Policy Optimization (GRPO) as our reinforcement learning algorithm. We fine-tune the SimpleAR-0.5B-SFT model for one epoch using 10K captions generated by GENERATE ANY SCENE, each paired with their scene graph-derived QA sets. For reward evaluation, we use Qwen2.5-VL-3B, a lightweight open-source vision-language model, to answer these QA pairs given the model-generated images. The reward is computed as the accuracy across all questions. This fine-grained, scene graph-aligned reward provides precise feedback on compositional faithfulness. As a baseline, we compare against SimpleAR-0.5B-RL, trained with CLIP-based rewards on 11K captions from real world datasets for one epoch. We evaluate our scene graph-based reward model on three benchmarks: DPG-Bench [10], GenEval [9], and GenAI-Bench [38]. More details are in Appendix E.

**GENERATE ANY SCENE rewards outperform CLIP.** As shown in Table 2, our method outperforms both SFT and CLIP-RL models and achieves a significant improvement, demonstrating superior

compositional faithfulness driven by explicit scene graph rewards. Importantly, this performance gain is directly enabled by the GENERATE ANY SCENE engine, which constructs explicit scene graphs to generate compositional captions. GENERATE ANY SCENE provides a structured and cognitively aligned visual representation, from which we derive exhaustive QA pairs with minimal additional cost. Combined with lightweight VLM judge, this approach offers a scalable, low-cost solution for semantic-level reward modeling.

Table 2: Evaluation on the DPG, GenEval and GenAI benchmark. GRPO training with our reward model outperforms both SFT baseline and CLIP-RL models. TO: two objects, P: position, CA: color attribute.

| Method | DPG-Bench | | | GenEval | | | | GenAI-Bench | | |
|---|---|---|---|---|---|---|---|---|---|---|
| | Global | Relation | Overall | TO | P | CA | Overall | Basic | Advanced | All |
| SimpleAR-0.5B-SFT | 85.02 | 86.59 | 78.48 | 0.73 | 0.22 | 0.23 | 0.53 | 0.74 | 0.60 | 0.66 |
| SimpleAR-0.5B-RL (Clip) | 86.64 | 88.51 | 79.66 | **0.82** | 0.26 | **0.38** | 0.59 | **0.75** | 0.60 | 0.67 |
| **SimpleAR-0.5B-RL (Ours)** | **88.46** | **90.13** | **80.50** | 0.81 | **0.31** | **0.38** | **0.61** | **0.75** | **0.61** | **0.68** |

# 6 Improving generated-content detection

Advances in *Text-to-Vision generation* underscore the need for effective content moderation [43]. Major challenges include the lack of high-quality and diverse datasets and the difficulty of generalizing detection across models *Text-to-Vision generation* [44, 45]. GENERATE ANY SCENE addresses these issues by enabling scalable, systematical generation of compositional captions, increasing the diversity and volume of synthetic data. This approach enhances existing datasets by compensating for their limited scope-from realistic to imaginative-and variability.

**Experiment setup.** To demonstrate GENERATE ANY SCENE's effectiveness in training generated content detectors, we used the $D^3$ dataset [46] as a baseline. We sampled 5K captioned real and SDv1.4-generated image pairs from $D^3$ and generated 5K additional images with GENERATE ANY SCENE captions. We trained a ViT–T [47] model with a single-layer linear classifier, and compared models trained with samples solely from $D^3$ against those trained with samples GENERATE ANY SCENE and $D^3$.

**GENERATE ANY SCENE improves generated content detectors.** We evaluate the detector's generalization on the GenImage [48] validation set and images generated using GENERATE ANY SCENE captions. Figure 6 demonstrates that combining GENERATE ANY SCENE-generated images with real-world captioned images consistently enhances detection performance, particularly across cross-model scenarios and diverse visual scenes. More details are in Appendix F.

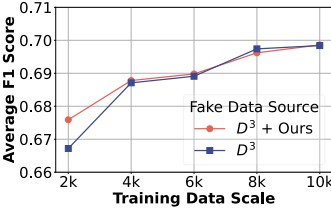 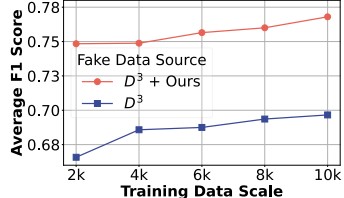 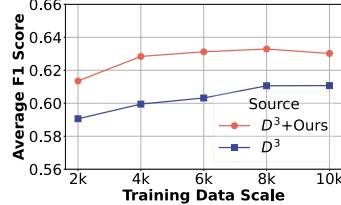

(a) **In-domain testing (Same Model - SD v1.4)**: Detection results on images generated by SD v1.4 using the GenImage dataset.

(b) **In domain testing (cross-model)**:Average detection results on images generated by multiple models using our captions.

(c) **Out of domain**: Average detection results on images generated by multiple models using captions from the GenImage dataset.

Figure 6: **Results for Application 4: Generated content detector**. Comparison of detection performance across different data scales using $D^3$ alone versus the combined $D^3$ + GENERATE ANY SCENE training set in cross-model and cross-dataset scenarios.

# 7 Comprehensive evaluation with GENERATE ANY SCENE

We conduct extensive evaluations of text-to-vision models using GENERATE ANY SCENE. Specifically, we synthesize 10K captions for text-to-image, 10K for text-to-video, and 1K for text-to-3D,

covering diverse scene structures and content topics. We evaluate 12 text-to-image, 9 text-to-video, and 5 text-to-3D models. Evaluations combine GENERATE ANY SCENE synthetic scene graphs with existing metrics (e.g., CLIP Score [49], VQA Score [37], TIFA Score [23, 31]) to assess semantic similarity, faithfulness, and human preference alignment. Our key findings include: (1) DiT-backbone text-to-image models align more closely with input captions than UNet-backbone models. (2) Text-to-video models struggle with balancing dynamics and consistency, while both text-to-video and text-to-3D models show notable gaps in human preference alignment. Additionally, we leverage GENERATE ANY SCENE's controllable caption generation to conduct fine-grained evaluations. These analyses cover varying levels of perplexity, scene complexity, and commonsense reasoning, as well as performance across different content categories. Details are in Appendix A.

## 8    Related work

***Text-to-Vision generation* models.**    *Text-to-Image generation* advances are driven by diffusion models and LLMs. Some open-source models [22, 50, 51, 52, 53, 54] use UNet backbones to refine images iteratively. In parallel, Diffusion Transformers (DiTs) architectures[55, 56, 57, 58] have emerged as a better alternative in capturing long-range dependencies and improving coherence. Proprietary models like DALL-E 3 [3] and Imagen 3 [59] still set the state-of-the-art. Based on *Text-to-Image generation* method, *Text-to-Video generation* models typically utilize time-aware architectures to ensure temporal coherence across frames [60, 61, 62, 63, 64, 65, 66, 67]. In *Text-to-3D generation*, recent proposed models [4, 68, 69, 70, 71] integrate the diffusion models with Neural Radiance Fields (NeRF) rendering to generate diverse 3D objects. Recent studies [26, 42, 72, 73] have also explored the integration of image generation into a unified multimodal language model (MLM) framework based on auto-regressive transformer architectures, demonstrating promising improvements in both performance and efficiency.

**Synthetic captions for *Text-to-Vision generation*.**    Captions for *Text-to-Vision generation* models vary greatly in diversity, complexity, and compositionality. This variation makes it challenging and costly to collect large-scale and diverse captions written by humans. Consequently, synthetic captions have been widely used for both training [74, 39, 75, 76, 8, 77, 78, 79] and evaluation purposes [7]. For example, training methods like LLM-Grounded Diffusion [74] leverage LLM-generated captions to enhance the model's understanding and alignment with human instruction. For evaluation, benchmarks such as T2I-CompBench [7] and T2V-CompBench [8] utilize benchmarks generated by LLMs. However, LLMs are hard to control and may introduce exhibit systematic bias. In this work, we propose a programmatic scene graph-based data engine that can generate infinitely diverse captions for improving *Text-to-Vision generation* models.

**Finetuning techniques for *Text-to-Vision generation*.**    To accommodate the diverse applications and personalization needs in text-to-vision models, numerous fine-tuning techniques have been developed. LoRA [40] reduces fine-tuning costs via low-rank weight updates, while Textual Inversion [80, 81] introduces new word embeddings for novel concepts without altering core parameters. DreamBooth [82] adapts models to specific subjects or styles using a few personalized images, and DreamSync [39] enables models to self-improve by learning from their own high-quality outputs. Recently, RLHF [26, 41, 42] in *Text-to-Vision generation* has shown promise as an efficient fine-tuning strategy. In this work, we use several fine-tuning techniques with GENERATE ANY SCENE to improve *Text-to-Vision generation* models.

## 9    Conclusion

We present GENERATE ANY SCENE, a system leveraging scene graph programming to generate diverse and compositional synthetic captions for *Text-to-Vision generation* tasks. It extends beyond existing real-world caption datasets to include comprehensive scenes and even implausible scenarios. To demonstrate the effectiveness of GENERATE ANY SCENE, we explore four applications: (1) self-improvement by iteratively optimizing models, (2) distillation of proprietary model strengths into open-source models, (3) a scene-graph-based efficient reward model within the GRPO, and (4) robust content moderation with diverse synthetic data. GENERATE ANY SCENE highlights the importance of synthetic data in improving *Text-to-Vision generation*, and addresses the need to systematically define and scalably produce the space of visual scenes.

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
