# A Evaluating *Text-to-Vision generation* models with GENERATE ANY SCENE

## A.1 Experiment Settings

**Models.** We conduct experiments on 12 *Text-to-image* models [54, 50, 22, 51, 52, 55, 56, 57, 58, 3], 9 *Text-to-Video* models [63, 83, 62, 60, 61, 64, 67, 66, 65], and 5 *Text-to-3D* models [68, 71, 69, 4, 70].

- For *Text-to-Image generation*, we select a range of open-source models, including those utilizing UNet backbones, such as *DeepFloyd IF* [54], *SDv2.1* [22], *SDXL* [50], *Playground v2.5* [51], and *Wuerstchen v2* [52], as well as models with DiT backbones, including *SD3 Medium* [55], *PixArt-α* [56], *PixArt-Σ* [57], *FLUX.1-schnell* [58], *FLUX.1-dev* [58], and FLUX 1. Closed-source models, such as *DaLL-E 3* [3] and *FLUX1.1 PRO* [58], are also assessed to ensure a comprehensive comparison. All models are evaluated at a resolution of $1024 \times 1024$ pixels.

- For *Text-to-Video generation*, we select nine open-source models: *ModelScope* [63], *ZeroScope* [83], *Text2Video-Zero* [62], *CogVideoX-2B* [66], *VideoCrafter2* [65], *AnimateLCM* [61], *AnimateDiff* [60], *FreeInit* [64], and *Open-Sora 1.2* [67]. We standardize the frame length to 16 across all video models for fair comparisons.

- For *Text-to-3D generation*, we evaluate five recently proposed models: *SJC* [69], *Dream-Fusion* [68], *Magic3D* [71], *Latent-NeRF* [70], and *ProlificDreamer* [4]. We employ the implementation and configurations provided by ThreeStudio [84] and generate videos by rendering from 120 viewpoints. To accelerate inference, we omit the refinement stage. For *Magic3D* and *DreamFusion*, we respectively use *DeepFloyd IF* and *SDv2.1* as their 2D backbones.

**Metrics.** Across all *Text-to-Vision generation* tasks, we use *Clip Score* [49] (semantic similarity), *VQA Score* [37] (faithfulness), *TIFA Score* [23, 31] (faithfulness), *Pick Score* [85] (human preference), and *ImageReward Score* [86] (human preference) as general metrics:

- *Clip Score*: Assesses semantic similarity between images and text.

- *VQA Score* and *TIFA Score*: Evaluate faithfulness by generating question-answer pairs and measuring answer accuracy from images.

- *Pick Score* and *ImageReward Score*: Capture human preference tendencies.

We also use metrics in VBench [87] to evaluate *Text-to-Video generation* models on fine-grained dimensions, such as consistency and dynamics, providing detailed insights into video performance.

For *Text-to-Video generation* and *Text-to-3D generation* tasks:

- We calculate *Clip Score*, *Pick Score*, and *ImageReward Score* on each frame, then average these scores across all frames to obtain an overall video score.

- For *VQA Score* and *TIFA Score*, we handle *Text-to-Video generation* and *Text-to-3D generation* tasks differently:

  ○ In *Text-to-Video generation* tasks, we uniformly sample four frames from the 16-frame sequence and arrange them in a $2 \times 2$ grid image.

  ○ For *Text-to-3D generation* tasks, we render images at 45-degree intervals from nine different viewpoints and arrange them in a $3 \times 3$ grid.

This sampling approach optimizes inference speed without affecting score accuracy [37].

**Synthetic captions.** We evaluate our *Text-to-Image generation* and *Text-to-Video generation* models on 10K randomly generated captions, with scene graph complexity ranging from 3 to 12 and scene attributes from 0 to 5, using unrestricted metadata. The captions exhibit an average graph degree of 1.15, with values spanning from 0.0 to 0.8. The mean number of connected components per scene graph is 3.51, ranging from 1 to 11. For *Text-to-3D generation* models, due to their limitations in handling complex captions and time-intensive generation, we restrict scene graph complexity to 1-3, scene attributes to 0-2, and evaluate on 1K captions.

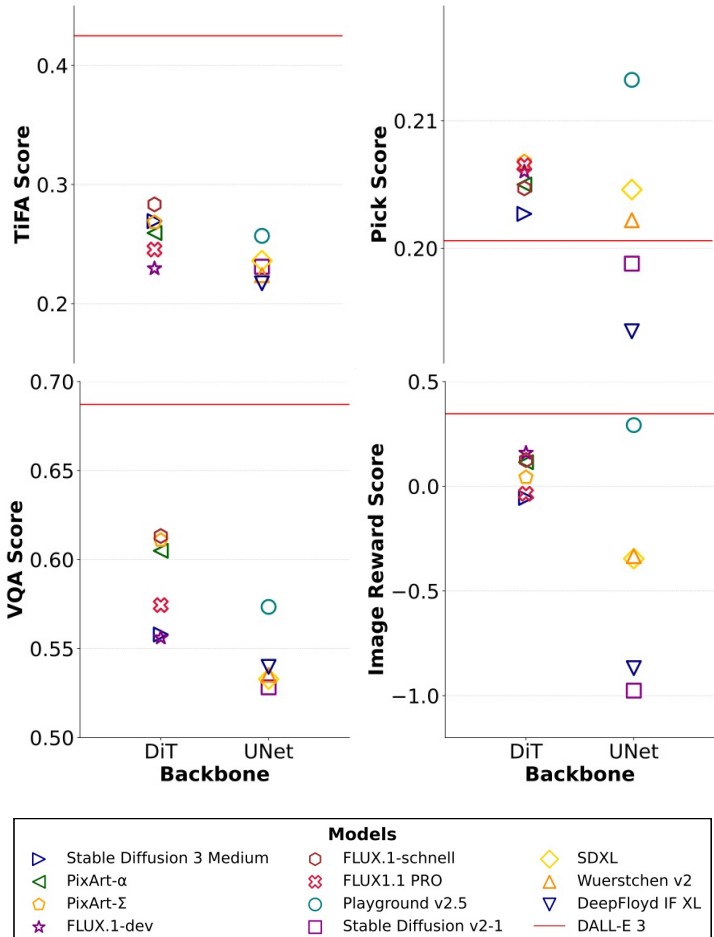

Figure 7: Comparative evaluation of *Text-to-Image generation* models across different backbones (DiT and UNet) using multiple metrics: *TIFA Score*, *Pick Score*, *VQA Score*, and *ImageReward Score*.

## A.2 Overall results

We evaluate *Text-to-Image generation*, *Text-to-Video generation*, and *Text-to-3D generation* models on GENERATE ANY SCENE.

Table 3: Overall performance of *Text-to-Image generation* models over 10K GENERATE ANY SCENE captions. †Evaluated on a 1K caption subset due to inference cost constraints.

| Model | clip score | pick score | vqa score | tifa score | image reward score |
|---|---|---|---|---|---|
| Playground v2.5 [51] | 0.2581 | 0.2132 | 0.5734 | 0.2569 | 0.2919 |
| Stable Diffusion v2-1 [22] | 0.2453 | 0.1988 | 0.5282 | 0.2310 | -0.9760 |
| SDXL [50] | 0.2614 | 0.2046 | 0.5328 | 0.2361 | -0.3463 |
| Wuerstchen v2 [52] | 0.2448 | 0.2022 | 0.5352 | 0.2239 | -0.3339 |
| DeepFloyd IF XL [54] | 0.2396 | 0.1935 | 0.5397 | 0.2171 | -0.8687 |
| Stable Diffusion 3 Medium [55] | 0.2527 | 0.2027 | 0.5579 | 0.2693 | -0.0557 |
| PixArt-$\alpha$ [56] | 0.2363 | 0.2050 | 0.6049 | 0.2593 | 0.1149 |
| PixArt-$\Sigma$ [57] | 0.2390 | 0.2068 | 0.6109 | 0.2683 | 0.0425 |
| FLUX.1-dev [58] | 0.2341 | 0.2060 | 0.5561 | 0.2295 | 0.1588 |
| FLUX.1-schnell [58] | 0.2542 | 0.2047 | 0.6132 | 0.2833 | 0.1251 |
| FLUX1.1 PRO [58]† | 0.2315 | 0.2065 | 0.5744 | 0.2454 | -0.0361 |
| Dalle-3 [3] | 0.2518 | 0.2006 | 0.6871 | 0.4249 | 0.3464 |

***Text-to-Image generation* results. (Figure 7, Table 3)**

1. DiT-backbone models outperform UNet-backbone models on *VQA Score* and *TIFA Score*, indicating greater faithfulness and comprehensiveness to input captions.

2. Despite using a UNet architecture, *Playground v2.5* achieves higher *Pick Score* and *ImageReward Score* scores than other open-source models. We attribute this to *Playground v2.5* 's alignment with human preferences achieved during training.

3. The closed-source model *DaLL-E 3* maintains a significant lead in *VQA Score*, *TIFA Score*, and *ImageReward Score*, demonstrating strong faithfulness and alignment with captions across generated content.

*Text-to-Video generation* **results. (Table 4,5)**

Table 4: Overall performance of open-source *Text-to-Video generation* models over 10K GENERATE ANY SCENE captions. Red Cell is the highest score. Yellow Cell is the second highest score.[†]Close-source models are evaluated on a 1K caption subset due to high inference cost.

| Model | clip score | pick score | image reward score | VQA score | TiFA score |
|---|---|---|---|---|---|
| VideoCraft2 [65] | 0.2398 | 0.1976 | -0.4202 | 0.5018 | 0.2466 |
| AnimateLCM [61] | 0.2450 | 0.1987 | -0.5754 | 0.4816 | 0.2176 |
| AnimateDiff [60] | 0.2610 | 0.1959 | -0.7301 | 0.5255 | 0.2208 |
| Open-Sora 1.2 [67] | 0.2259 | 0.1928 | -0.6277 | 0.5519 | 0.2414 |
| FreeInit [64] | 0.2579 | 0.1950 | -0.9335 | 0.5123 | 0.2047 |
| ModelScope [63] | 0.2041 | 0.1886 | -1.9172 | 0.3840 | 0.1219 |
| Text2Video-Zero [62] | 0.2539 | 0.1933 | -1.2050 | 0.4753 | 0.1952 |
| CogVideoX-2B [66] | 0.2038 | 0.1901 | -1.2301 | 0.4585 | 0.1997 |
| ZeroScope [83] | 0.2289 | 0.1933 | -1.1599 | 0.4892 | 0.2388 |
| KLING 1.6 [88][†] | 0.2215 | 0.1985 | -0.3419 | 0.5307 | 0.2802 |
| Wanx 2.1 [89][†] | 0.2308 | 0.1969 | -0.1418 | 0.5970 | 0.3328 |

Table 5: Overall performance of open-source *Text-to-Video generation* models over 10K GENERATE ANY SCENE captions with VBench metrics. Red Cell is the highest score. Blue Cell is the lowest score.

| Model | subject consistency | background consistency | motion smoothness | dynamic degree | aesthetic quality | imaging quality |
|---|---|---|---|---|---|---|
| Open-Sora 1.2 | 0.9964 | 0.9907 | 0.9973 | 0.0044 | 0.5235 | 0.6648 |
| Text2Video-Zero | 0.8471 | 0.9030 | 0.8301 | 0.9999 | 0.4889 | 0.7018 |
| VideoCraft2 | 0.9768 | 0.9688 | 0.9833 | 0.3556 | 0.5515 | 0.6974 |
| AnimateDiff | 0.9823 | 0.9733 | 0.9859 | 0.1406 | 0.5427 | 0.5830 |
| FreeInit | 0.9581 | 0.9571 | 0.9752 | 0.4440 | 0.5200 | 0.5456 |
| ModelScope | 0.9795 | 0.9831 | 0.9803 | 0.1281 | 0.3993 | 0.6494 |
| AnimateLCM | 0.9883 | 0.9802 | 0.9887 | 0.0612 | 0.6323 | 0.6977 |
| CogVideoX-2B | 0.9583 | 0.9602 | 0.9823 | 0.4980 | 0.4607 | 0.6098 |
| ZeroScope | 0.9814 | 0.9811 | 0.9919 | 0.1670 | 0.4582 | 0.6782 |

1. Open-source text-to-video models face challenges in balancing dynamics and consistency (Table 5). This is especially evident in *Open-Sora 1.2*, which achieves high consistency but minimal dynamics, and *Text2Video-Zero*, which excels in dynamics but suffers from frame inconsistency.

2. All models exhibit negative *ImageReward Score* (Table 4), suggesting a lack of human-preferred visual appeal in the generated content, even in cases where certain models demonstrate strong semantic alignment.

3. As expected, SOTA close-source text-to-video models outperform others overall, particularly in image reward, VQA score, and TIFA score. This indicates their superior alignment with human preferences, as well as stronger faithfulness and compositional capabilities in generation.

4. Among open-source models, *VideoCrafter2* strikes a balance across key metrics, leading in human-preference alignment, faithfulness, consistency, and dynamic.

*Text-to-3D generation* **results. (Table 6)**

Table 6: Overall performance of *Text-to-3D generation* models over 1K GENERATE ANY SCENE captions. [†]Evaluated on a 100 caption subset due to high inference cost.

| Model | clip score | pick score | vqa score | tifa score | image reward score |
|---|---|---|---|---|---|
| Latent-NeRF [70] | 0.2115 | 0.1910 | 0.4767 | 0.2216 | -1.5311 |
| DreamFusion-sd [68] | 0.1961 | 0.1906 | 0.4421 | 0.1657 | -1.5582 |
| Magic3D-sd [71] | 0.1947 | 0.1903 | 0.4193 | 0.1537 | -1.6327 |
| SJC [69] | 0.2191 | 0.1915 | 0.5015 | 0.2563 | -1.4370 |
| DreamFusion-IF [68] | 0.1828 | 0.1857 | 0.3872 | 0.1416 | -1.9353 |
| Magic3D-IF [71] | 0.1919 | 0.1866 | 0.4039 | 0.1537 | -1.8465 |
| ProlificDreamer [4] | 0.2125 | **0.1940** | **0.5411** | 0.2704 | -1.2774 |
| Meshy-4 [90][†] | **0.2163** | 0.1922 | 0.5290 | **0.2908** | **-1.0496** |

1. Among open-source models, *ProlificDreamer* outperforms other models, particularly in *ImageReward Score*, *VQA Score* and *TIFA Score*.

2. All models receive negative *ImageReward Score* scores, highlighting a significant gap between human preference and current *Text-to-3D generation* generation capabilities.

3. Meshy-4 demonstrates overall superior performance compared to all open-source models, especially in terms of *Clip Score*, *TIFA Score* and *ImageReward Score*, reflecting its strengths in semantic generation and human preference alignment.

## A.3 More Analysis with GENERATE ANY SCENE

With GENERATE ANY SCENE, we can generate infinitely diverse and highly controllable captions. Using GENERATE ANY SCENE, we conduct several analyses to provide insights into the performance of today's *Text-to-Vision generation* models.

### A.3.1 Performance analysis across caption properties

In this section, we delve into how model performance varies with respect to distinct properties of GENERATE ANY SCENE captions. While GENERATE ANY SCENE is capable of generating an extensive diversity of captions, these outputs inherently differ in key characteristics that influence model evaluation. Specifically, we examine three properties of the caption: Commonsense, Perplexity, and Scene Graph Complexity (captured as the number of elements in the captions). These properties are critical in understanding how different models perform across a spectrum of linguistic and semantic challenges presented by captions with varying levels of coherence, plausibility, and compositional richness.

**Perplexity. (Figure 8)** Perplexity is a metric used to measure a language model's unpredictability or uncertainty in generating a text sequence. A higher perplexity value indicates that the sentences are less coherent or less likely to be generated by the model.

As shown in Figure 8, From left to right, when perplexity increases, indicating that the sentences become less reasonable and less typical of those generated by a language model, we observe no clear or consistent trends across all models and metrics. This suggests that the relationship between perplexity and model performance varies depending on the specific model and evaluation metric.

**Commonsense. (Figure 9)** Commonsense is an inherent property of text. We utilize the Vera Score [91], a metric generated by a fine-tuned LLM to evaluate the text's commonsense level.

As shown in Figure 9, from left to right, as the Vera Score increases—indicating that the captions exhibit greater commonsense reasoning—we observe a general improvement in performance across

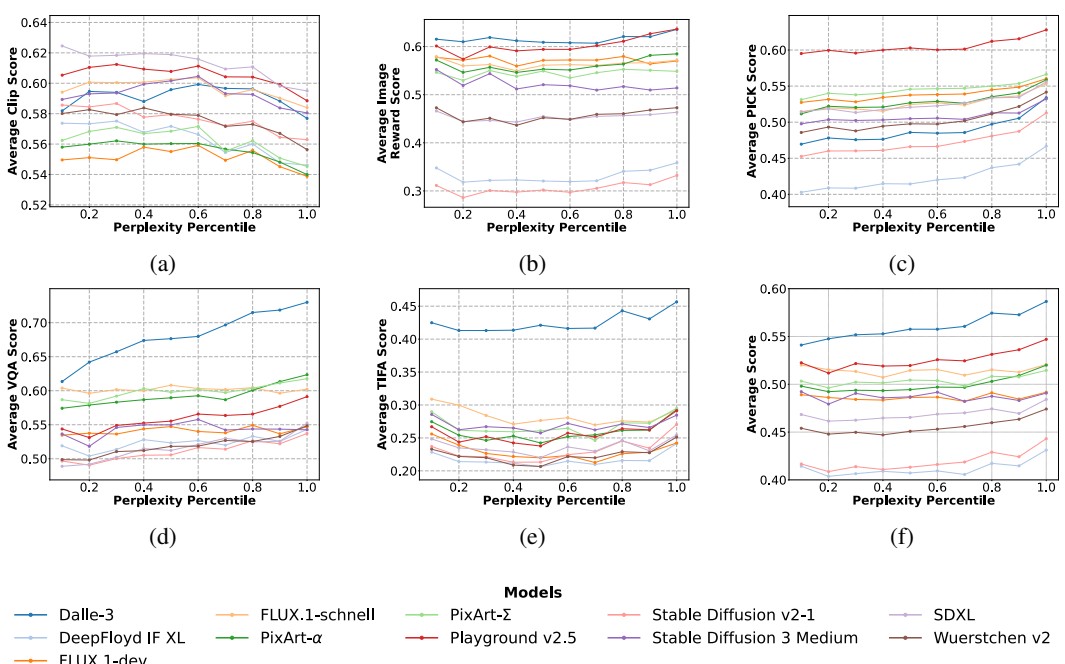

Figure 8: Average performance of models across different percentiles of perplexity of captions, evaluated on various metrics. From left to right, the perplexity decreases, indicating captions that are progressively more reasonable and easier for the LLM to generate.

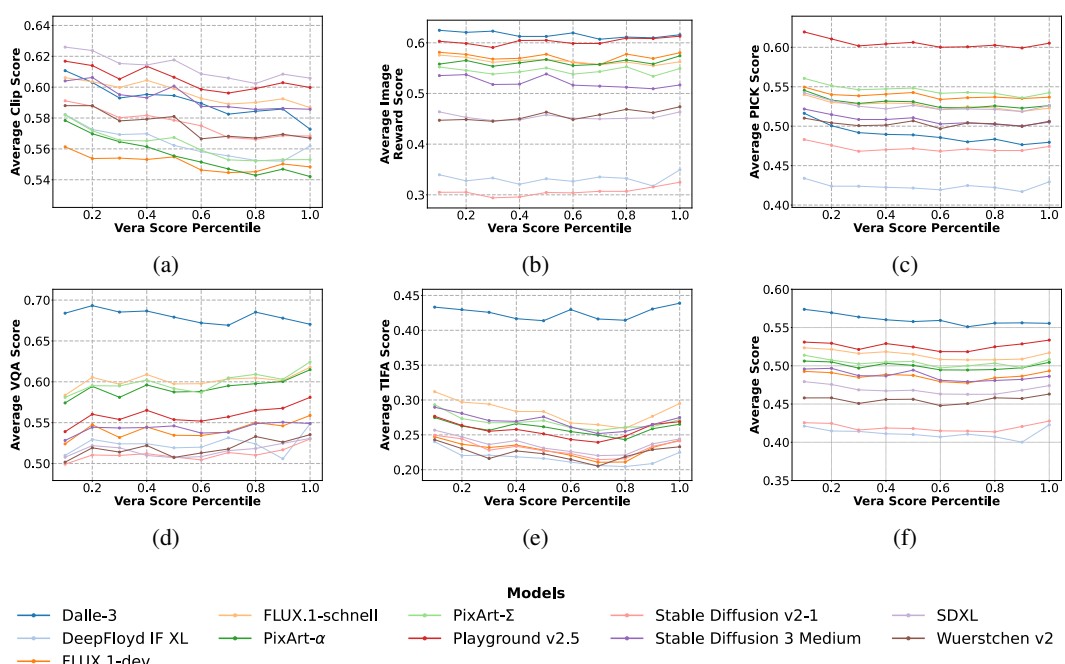

Figure 9: Average performance of models across different percentiles of Vera Score for captions, evaluated on various metrics. From left to right, the Vera Score decreases, indicating captions that exhibit less commonsense reasoning and are more likely to describe implausible scenes.

all metrics and models, except for *Clip Score*. This trend underscores the correlation between commonsense-rich captions and enhanced model performance.

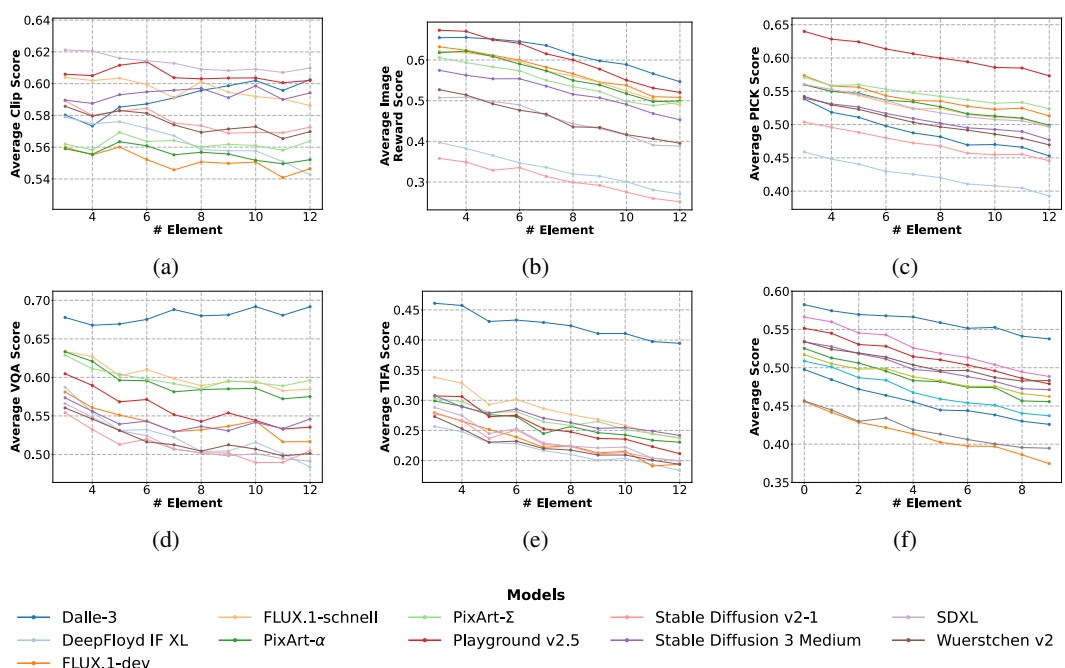

Figure 10: Average performance of models across different numbers of elements (objects + attributes + relations) in the scene graph (complexity of the scene graph) of the captions, evaluated on various metrics. From left to right, as the number of elements (complexity) increases, the scene graphs become more complicated and compositional.

**Element Numbers (Complexity of Scene Graph). (Figure 10)** Finally, we evaluate model performance across total element numbers in the captions, which represent the complexity of scene graphs (objects + attributes + relations).

From left to right, the complexity of scene graphs becomes higher, reflecting more compositional and intricate captions. Across most metrics and models, we observe a noticeable performance decline as the scene graphs become more complex. However, an interesting exception is observed in the performance of *DaLL-E 3*. Unlike other models, *DaLL-E 3* performs exceptionally well on *VQA Score* and *TIFA Score*, particularly on *VQA Score*, where it even shows a slight improvement as caption complexity increases. This suggests that *DaLL-E 3* may have a unique capacity to handle complex and compositional captions effectively.

### A.3.2 Analysis on different metrics

Compared with most LLM and VLM benchmarks that use multiple-choice questions and accuracy as metrics. There is no universal metric in evaluating *Text-to-Vision generation* models. Researchers commonly used model-based metrics like *Clip Score*, *VQA Score*, etc. Each of these metrics is created and fine-tuned for different purposes with bias. Therefore, we also analysis on different metrics.

*Clip Score* **isn't a universal metric.** *Clip Score* is one of the most widely used metrics in *Text-to-Vision generation* for evaluating the alignment between visual content and text. However, our analysis reveals that *Clip Score* is not a perfect metric and displays some unusual trends. For instance, as shown in Figures 8, 9, and 10, we compute the perplexity across 10K captions used in our study, where higher perplexity indicates more unpredictable or disorganized text. Interestingly, unlike other metrics, *Clip Score* decreases as perplexity lowers, suggesting that *Clip Score* tends to favor more disorganized text. This behavior is counterintuitive and highlights the potential limitations of using *Clip Score* as a robust alignment metric.

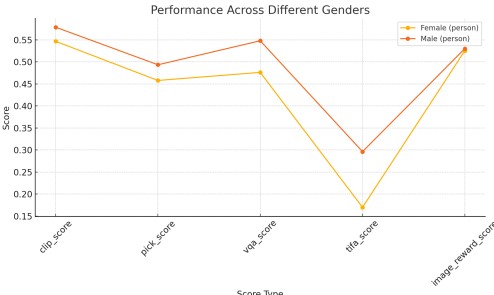
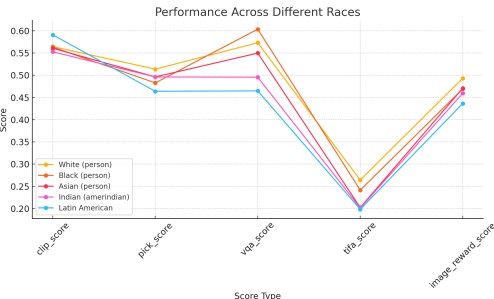

Figure 11: Average performance scores of all models across different genders evaluated using various metrics.

Figure 12: Average performance scores of all models across different races evaluated using various metrics.

**Limitations of human preference-based metrics.** We use two metrics fine-tuned using human preference data: *Pick Score* and *ImageReward Score*. However, we found that these metrics exhibit a strong bias toward the data on which they were fine-tuned. For instance, as shown in Table 3, *Pick Score* assigns similar scores across all models, failing to provide significant differentiation or meaningful insights into model performance. In contrast, *ImageReward Score* demonstrates clearer preferences, favoring models such as *DaLL-E 3* and *Playground v2.5*, which incorporated human-alignment techniques during their training. However, this metric shows a significant drawback: it assigns disproportionately large negative scores to models like *SDv2.1*, indicating a potential over-sensitivity to alignment mismatches. Such behavior highlights the limitations of these metrics in providing fair and unbiased evaluations across diverse model architectures.

*VQA Score* **and** *TIFA Score* **are relative reliable metrics.** Among the evaluated metrics, *VQA Score* and *TIFA Score* stand out by assessing model performance on VQA tasks, rather than relying solely on subjective human preferences. This approach enhances the interpretability of the evaluation process. Additionally, we observed that the results from *VQA Score* and *TIFA Score* show a stronger correlation with other established benchmarks. Based on these advantages, we recommend prioritizing these two metrics for evaluation. However, it is important to note that their effectiveness is constrained by the limitations of the VQA models utilized in the evaluation.

### A.3.3 Fairness analysis

We evaluate fairness by examining the model's performance across different genders and races. Specifically, we calculate the average performance for each node and its associated child nodes within the taxonomy tree constructed for objects. For example, the node "females" includes child nodes such as "waitresses," and their combined performance is considered in the analysis.

**Gender.** In gender, we observe a notable performance gap between females and males, as could be seen from Figure 11, Models are better at generating male concepts.

**Race.** There are also performance gaps in different races. From Figure 12, we found that "white (person)" and "black (person)" perform better than "asian (person)", "Indian (amerindian)", and "Latin American".

### A.3.4 Correlation of GENERATE ANY SCENE with other *Text-to-Vision generation benchmarks*

The GENERATE ANY SCENE benchmark uniquely relies entirely on synthetic captions to evaluate models. To assess the transferability of these synthetic captions, we analyzed the consistency in model rankings across different benchmarks [79, 38, 92]. Specifically, we identified the overlap of models evaluated by two benchmarks and computed the Spearman correlation coefficient between their rankings.

As shown in the figure 13, GENERATE ANY SCENE demonstrates a strong correlation with other benchmarks, such as Conceptmix [79] and GenAI Bench [38], indicating the robustness and reliability

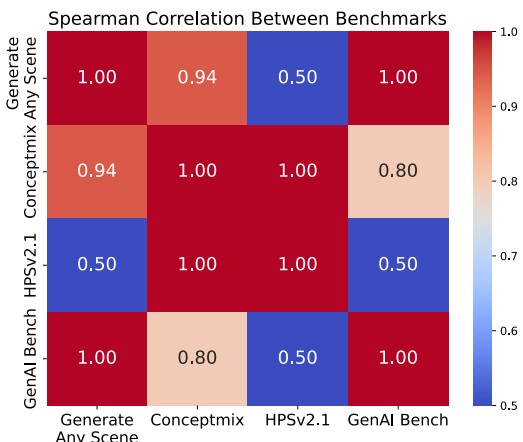

Figure 13: Correlation of GENERATE ANY SCENE with other popular *Text-to-Vision generation* benchmarks.

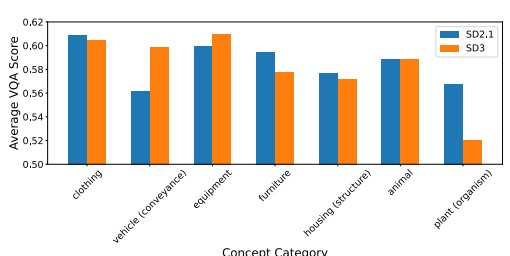

(a) *SDv2.1* vs. *SD3 Medium* on average *VQA Score* in fine-grained categories.

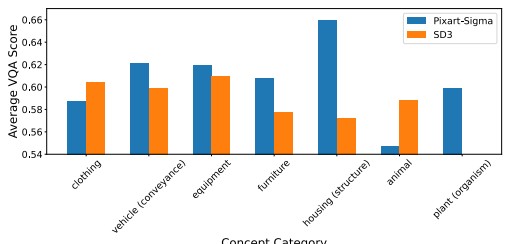

(b) *PixArt-Σ* vs. *SD3 Medium* on average *VQA Score* in fine-grained categories.

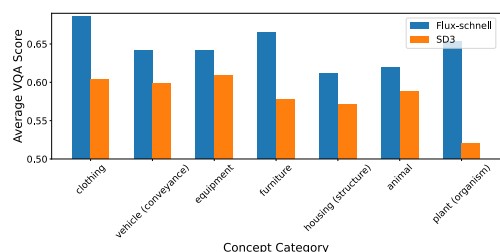

(c) *FLUX.1-schnell* vs. *SD3 Medium* on average *VQA Score* in fine-grained categories.

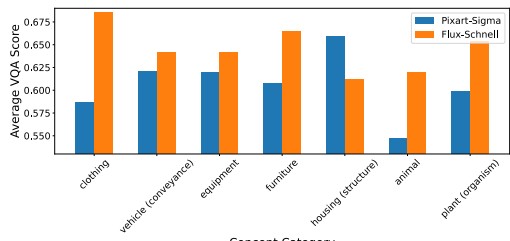

(d) *PixArt-Σ* vs. *FLUX.1-schnell* on average *VQA Score* in fine-grained categories.

Figure 14: Pairwise comparison on average *VQA Score* in fine-grained categories.

of GENERATE ANY SCENE's synthetic caption-based evaluations. This suggests that the synthetic captions generated by GENERATE ANY SCENE can effectively reflect model performance trends, aligning closely with those observed in benchmarks using real-world captions or alternative evaluation methods.

### A.3.5 Case study: Pairwise fine-grained model comparison

Evaluating models using a single numerical average score can be limiting, as different training data often lead models to excel in generating different types of concepts. By leveraging the taxonomy we developed for GENERATE ANY SCENE, we can systematically organize these concepts and evaluate each model's performance on specific concepts over the taxonomy. This approach enables a more detailed comparison of how well models perform on individual concepts rather than relying solely on an overall average score. Our analysis revealed that, while the models may achieve similar average

performance, their strengths and weaknesses vary significantly across different concepts. Here we present a pairwise comparison of models across different metrics.

## B Details of Taxonomy of Visual Concepts

To construct a scene graph, we utilize three primary types of metadata: objects, attributes, and relations, which represent the structure of a visual scene. Additionally, scene attributes—which include factors like image style, perspective, and video time span—capture broader aspects of the visual content. Together, the scene graph and scene attributes form a comprehensive representation of the scene.

Our metadata is further organized using a well-defined taxonomy, enhancing the ability to generate controllable captions. This hierarchical taxonomy not only facilitates the creation of diverse scene graphs, but also enables fine-grained and systematic model evaluation.

**Objects.** To enhance the comprehensiveness and taxonomy of object data, we leverage noun synsets and the structure of WordNet [32]. In WordNet, a *physical object* is defined as *"a tangible and visible entity; an entity that can cast a shadow."* Following this definition, we designate the *physical object* as the root node, constructing a hierarchical tree with all *28,787* hyponyms under this category as the set of objects in our model.

Following WordNet's hypernym-hyponym relationships, we establish a tree structure, linking each object to its primary parent node based on its first-listed hypernym. For objects with multiple hypernyms, we retain only the primary parent to simplify the hierarchy. Furthermore, to reduce ambiguity, if multiple senses of a term share the same parent, we exclude that term itself and reassign its children to the original parent node. This approach yields a well-defined and disambiguated taxonomy.

**Attributes.** The attributes of a scene graph represent properties or characteristics associated with each object. We classify these attributes into *nine* primary categories. For *color*, we aggregate *677* unique entries sourced from Wikipedia [33]. The *material* category comprises *76* types, referenced from several public datasets [93, 94, 95]. The *texture* category includes *42* kinds from the Describable Textures Dataset [96], while the *architectural style* encompasses *25* distinct styles [97]. Additionally, we collect *85 states*, *41 shapes*, and *24 sizes*. For *human descriptors*, we compile 59 terms across subcategories, including body type and height. Finally, we collect *465* common *adjectives* covering general characteristics of objects to enhance the descriptive richness of our scene graphs.

**Relationships.** We leverage the Robin dataset [34] as the foundation for relationship metadata, encompassing six key categories: spatial, functional, interactional, social, emotional, and symbolic. With 10,492 relationships, the dataset provides a comprehensive and systematic repository that supports modeling diverse and complex object interactions. Its extensive coverage captures both tangible and abstract connections, forming a robust framework for accurate scene graph representation.

**Scene Attributes.** In *Text-to-Vision generation* tasks, people mainly focus on creating realistic images and art from a text description [98, 2, 3]. For artistic styles, we define scene attributes using *76* renowned *artists*, *41 genres*, and *126 painting styles* from WikiArt [99], along with *29* common *painting techniques*. For realistic imagery, we construct camera settings attributes across 6 categories: camera models, focal lengths, perspectives, apertures, depths of field, and shot scales. The camera models are sourced from the 1000 Cameras Dataset [100], while the remaining categories are constructed based on photography knowledge and common captions in *Text-to-Vision generation* tasks [1, 101]. To control scene settings, we categorize location, weather and lighting attributes, using 430 diverse locations from Places365 [35], alongside *76 weathers* and *57 lighting conditions*. For video generation, we introduce attributes that describe dynamic elements. These include 12 types of camera rig, 30 distinct camera movements, 15 video editing styles, and 27 temporal spans. The comprehensive scene attributes that we construct allow for the detailed and programmatic *Text-to-Vision generation* generation.

# C  Details of self-improving models with synthetic captions (Section 3)

## C.1  Experiment details

### C.1.1  Captions Preparation

To evaluate the effectiveness of our iterative self-improving *Text-to-Vision generation* model, we generated three distinct sets of 10K captions using GENERATE ANY SCENE, covering a sample complexity range from 3 to 12. These captions were programmatically created to reflect a spectrum of structured scene graph compositions, designed to challenge and enrich the model's learning capabilities.

For comparative analysis, we leveraged the Conceptual Captions (CC3M) [102] dataset, a large-scale benchmark containing approximately 3.3 million image-caption pairs sourced from web alt-text descriptions. CC3M is renowned for its diverse visual content and natural language expressions, encompassing a wide range of styles, contexts, and semantic nuances.

To ensure fair comparison, we randomly sampled three subsets of 10K captions from the CC3M dataset, matching the GENERATE ANY SCENE-generated caption sets in size. This approach standardizes data volume while enabling direct performance evaluation. The diversity and semantic richness of the CC3M captions serve as a robust benchmark to assess whether GENERATE ANY SCENE-generated captions can match or exceed the descriptive quality of real-world data across varied visual contexts.

### C.1.2  Dataset Construction and Selection Strategies

For the captions generated by GENERATE ANY SCENE, we employed a top-scoring selection strategy to construct the fine-tuning training dataset, using a random selection strategy as a baseline for comparison. Specifically, for each caption, the model generated eight images. Under the top-scoring strategy, we evaluated the generated images using the VQA score and selected the highest-scoring image as the best representation of the caption. This process yielded 10K top-ranked images per iteration, from which the top 25% (approximately 2.5k images) with the highest VQA scores were selected to form the fine-tuning dataset.

In the random selection strategy, one image was randomly chosen from the eight generated per caption, and 25% of these 10K randomly selected images were sampled to create the fine-tuning dataset, maintaining parity in data size.

For the CC3M dataset, each caption was uniquely paired with a real image. From the 10K real image-caption pairs sampled from CC3M, the top 25% with the highest VQA scores were selected as the fine-tuning training dataset. This ensured consistency in data size and selection criteria across all methods, facilitating a rigorous and equitable comparison of fine-tuning strategies.

### C.1.3  Fine-tuning details

We fine-tuned the *SDv1.5* using the LoRA technique. The training was conducted with a resolution of $512 \times 512$ for input images and a batch size of 8. Gradients were accumulated over two steps. The optimization process utilized the AdamW optimizer with $\beta_1 = 0.9$, $\beta_2 = 0.999$, an $\epsilon$ value of $1 \times 10^{-8}$, and a weight decay of $10^{-2}$. The learning rate was set to $1 \times 10^{-4}$ and followed a cosine scheduler for smooth decay during training. To ensure stability, a gradient clipping threshold of 1.0 was applied. The fine-tuning process was executed for one epoch, with a maximum of 2500 training steps. For the LoRA-specific configurations, we set the rank of the low-rank adaptation layers and the scaling factor $\alpha$ to be 128.

After completing fine-tuning for each epoch, we set the LoRA weight to 0.75 and integrate it into *SDv1.5* to guide image generation and selection for the next subset. For the CC3M dataset, images from the subsequent subset are directly selected.

In the following epoch, the fine-tuned LoRA parameters from the previous epoch are loaded and used to resume training on the current subset, ensuring continuity and leveraging the incremental improvements from prior iterations.

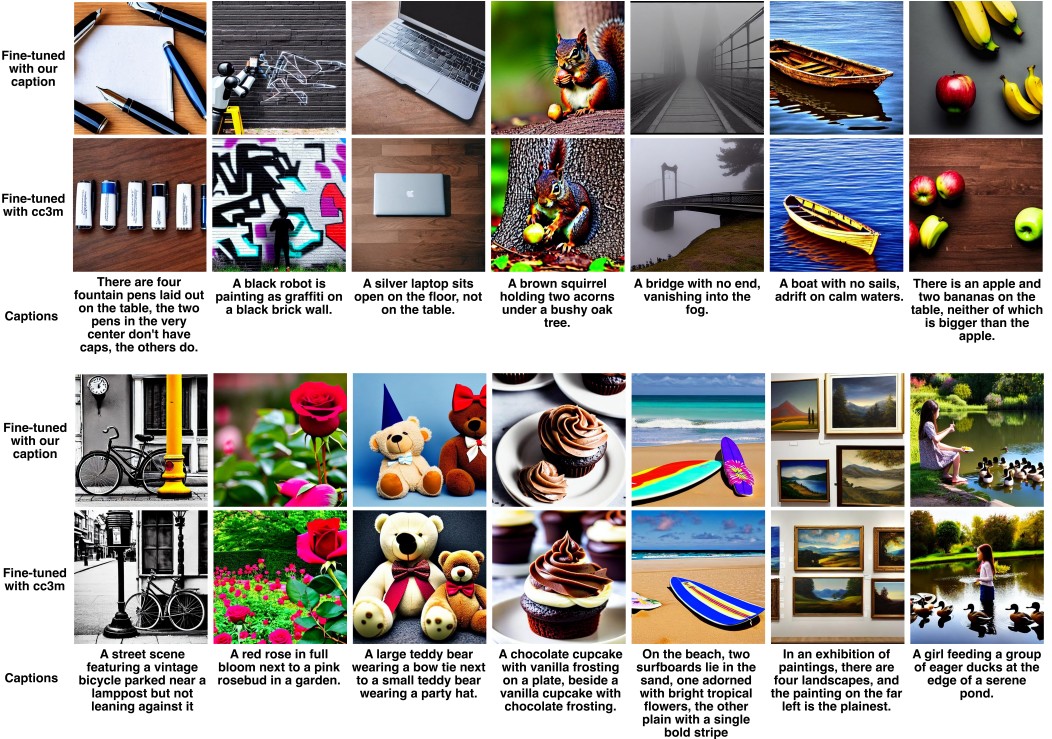

Figure 15: **Visualization of Different Caption Fine-Tuning.**

In Figure 15, we present results using our captions and the CC3M captions. The model fine-tuned with captions generated by GENERATE ANY SCENE demonstrates superior performance in terms of text semantic relevance and the generation of complex compositional scenes.

## C.2    More results of fine-tuning models

Aside from our own test set and GenAI benchmark, we also evaluated our fine-tuned *Text-to-Image generation* models on the Tifa Bench (Figure 16), where we observed the same trend: models fine-tuned with our captions consistently outperformed the original *SDv1.5* and CC3M fine-tuned models.

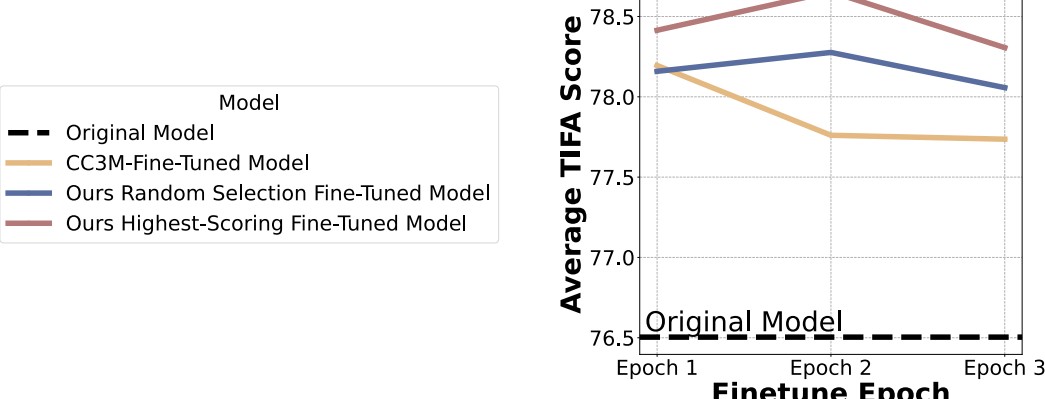

Figure 16: **Results for Application 1: Self-Improving Models**. Average TIFA score of *SDv1.5* fine-tuned with different data over TIFA Bench.

# D  Details of distilling targeted capabilities (Section 4)

## D.1  Collecting hard concepts

We selected 81 challenging object concepts where *SDv1.5* and *DaLL-E 3* exhibit the largest gap in *VQA Score*. To determine the score for each concept, we calculated the average VQA score of the captions containing that specific concept. The full list of hard concepts is shown below:

1. cloverleaf
2. aerie (habitation)
3. admixture
4. webbing (web)
5. platter
6. voussoir
7. hearthstone
8. puttee
9. biretta
10. yarmulke
11. surplice
12. overcoat
13. needlepoint
14. headshot
15. photomicrograph
16. lavaliere
17. crepe
18. tureen
19. bale
20. jetliner
21. square-rigger
22. supertanker
23. pocketcomb
24. filament (wire)
25. inverter
26. denture
27. lidar
28. volumeter
29. colonoscope
30. synchrocyclotron
31. miller (shaper)
32. alternator
33. dicer
34. trundle
35. paddle (blade)
36. harmonica
37. piccolo
38. handrest

39. rundle

40. blowtorch

41. volleyball

42. tile (man)

43. shuttlecock

44. jigsaw

45. roaster (pan)

46. maze

47. belt (ammunition)

48. gaddi

49. drawer (container)

50. tenter

51. pinnacle (steeple)

52. pegboard

53. afterdeck

54. scaffold

55. catheter

56. broomcorn

57. spearmint

58. okra (herb)

59. goatsfoot

60. peperomia

61. ammobium

62. gazania

63. echinocactus

64. birthwort

65. love-in-a-mist (passionflower)

66. ragwort

67. spicebush (allspice)

68. leadplant

69. barberry

70. hamelia

71. jimsonweed

72. undershrub

73. dogwood

74. butternut (walnut)

75. bayberry (tree)

76. lodestar

77. tapa (bark)

78. epicalyx

79. blackberry (berry)

80. stub

81. shag (tangle)

## D.2 Experiment details

We conducted targeted fine-tuning experiments on *SDv1.5* to evaluate GENERATE ANY SCENE's effectiveness in distilling model compositionality and learning hard concepts. For each task, we selected a dataset of 778 GENERATE ANY SCENE captions paired with images generated by *DaLL-E 3*. For compositionality, we selected multi-object captions from the existing dataset of 10K GENERATE ANY SCENE captions and paired them with the corresponding images generated by *DaLL-E 3*. To address hard concept learning, we first used *SDv1.5* to generate images based on the 10K GENERATE ANY SCENE captions and identified the hard concepts with the lowest VQA scores. These concepts were then used to create a subset of objects, which we recombined into our scene-graph based captions with complexity levels ranging from 3 to 9. Finally, we used *DaLL-E 3* to generate corresponding images for these newly composed captions.

The fine-tuning configurations were consistent with those used in the self-improving setup (Appendix C.1.3). To accommodate the reduced dataset size, the maximum training steps were set to 1000.

As a baseline, we randomly selected 778 images from 10K GENERATE ANY SCENE-generated images, using captions produced by GENERATE ANY SCENE. This ensured a controlled comparison between the targeted and random fine-tuning strategies.

# E  Details of reinforcement learning with a synthetic reward function (Section 5)

## E.1  Training data preparation

We adopt SimpleAR-0.5B-SFT [26] as our base model. Given that SImpleAR-0.5B-SFT is pretrained on high-quality real image datasets such as LAION [11] and CC3M [12], we aim to mitigate potential distributional shift between the original training data and the reinforcement learning phase. To this end, we perform metadata pre-selection for GENERATE ANY SCENE by analyzing the frequency of each object category appearing in the LAION dataset. Leveraging the controllable compositional capabilities of GENERATE ANY SCENE, we filter object categories by selecting the top 10% most frequent entries and constrain scene complexity to 3–6 objects per scene. Based on these conditions, we synthesize a set of 10K captions, ensuring semantic alignment with the base model's pretraining distribution while maintaining structural and content diversity.

## E.2  Experiment details

The detailed training configuration is provided in Table 7. We utilize $8 \times$ NVIDIA H100 GPUs (80GB HBM3), with one GPU allocated for online generation using vLLM. The total training time is approximately 14 hours.

Table 7: Scene-graph based GRPO Fine-tuning Configuration for SimpleAR

| Component | Details |
|---|---|
| Model Name | SimpleAR-0.5B-SFT |
| Model Size | $\sim$0.5B parameters |
| Training Policy | GRPO |
| Inference Engine | vLLM (GPU utilization = 0.7) |
| Completion Length | 4096 tokens |
| Training Epochs | 1 |
| Batch Size per Device | 4 |
| Learning Rate | $1 \times 10^{-5}$ |
| Scheduler | Cosine Annealing (min lr rate = 0.1) |
| Warm-up Ratio | 0.1 |
| Gradient Accumulation | 1 |

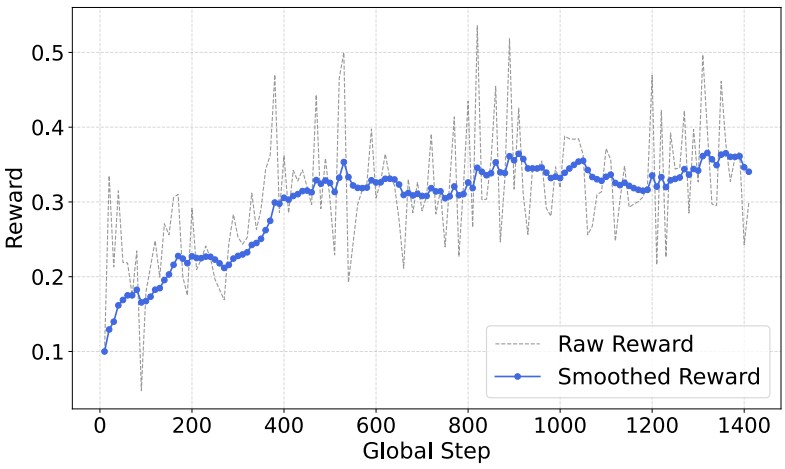

Figure 17: Reward progression during scene-graph based GRPO training.

Figure 17 illustrates the reward progression during training. A noticeable improvement in reward is observed following the application of a learning rate of 1e-5 combined with a warm-up strategy.

Overall, the reward increases by approximately 0.2, indicating effective learning under the adjusted training configuration.

In Table 2, we observe that the reproduced results of baseline models on DPG-Bench and GenEval Bench are slightly lower than those reported in the original paper. Considering the inherent stochasticity in generative model outputs, we cite the original results for comparison. For GenAI-Bench, all reported results are based on our own experimental evaluations.

# F Details of improving generated-content detection (Section 6)

## F.1 Experiment details

In this section, our goal is to validate that the more diverse captions generated by GENERATE ANY SCENE can complement existing datasets, which are predominantly composed of real-world images paired with captions. By doing so, we aim to train AI-generated content detectors to achieve greater robustness.

**Dataset preparation** We conducted comparative experiments between captions generated by GENERATE ANY SCENE and entries from the $D^3$ dataset. From the $D^3$ dataset, we randomly sampled 10K entries, each including a caption, a link to a real image, and an image generated by SD v1.4. Due to some broken links, we successfully downloaded 8.5K real images and retained 10K SD v1.4-generated images. We also used SD v1.4 to generate images based on 10K GENERATE ANY SCENE captions.

We varied the training data sizes based on the sampled dataset. Specifically, we sampled N real images from the 10K $D^3$ real images. For synthetic data, we compared N samples exclusively from $D^3$ with a mixed set of N/2 samples from 10K GENERATE ANY SCENE images and N/2 sampled from $D^3$, ensuring a total of N synthetic samples. Combined, this resulted in 2N training images. We tested 2N across various sizes, ranging from 2K to 10K.

**Detector architecture and training** We employed ViT-T [47] and ResNet-18 [103] as backbones for the detection models. Their pretrained parameters on ImageNet-21K were frozen, and the final classification head was replaced with a linear layer using a sigmoid activation function to predict the probability of an image being AI-generated. During training, We used Binary Cross-Entropy (BCE) as the loss function, and the AdamW optimizer was applied with a learning rate of $2e^{-3}$. Training was conducted with a batch size of 256 for up to 50 epochs, with early stopping triggered after six epochs of no improvement in validation performance.

**Testing** To evaluate the performance of models trained with varying dataset sizes and synthetic data combinations, we tested them on both GenImage and GENERATE ANY SCENE datasets to assess their in-domain and out-of-domain performance under different settings.

For GenImage, we used validation data from four models: SD v1.4, SD v1.5, MidJourney, and VQDM. Each validation set contained 8K real images and 8k generated images. For GENERATE ANY SCENE, we sampled 10K real images from CC3M and paired them with 10K generated images from each of the following models: *SDv2.1*, *PixArt-α*, *SD3 Medium*, and *Playground v2.5*. This created distinct test sets for evaluating model performance across different synthetic data sources.

Table 8: F1-Score Comparison of ResNet-18 and ViT-T Detectors Trained with $D^3$ and $D^3$+ GENERATE ANY SCENE Across In-Domain Settings

| Detector | Data Scale (2N) | SDv1.4 (In-domain, same model) | | SDv2.1 | | Pixart-α | | SDv3-medium | | Playground v2.5 | | Average (In-domain, cross model) | |
|---|---|---|---|---|---|---|---|---|---|---|---|---|---|
| | | $D^3$ + Ours | $D^3$ | $D^3$ + Ours | $D^3$ | $D^3$ + Ours | $D^3$ | $D^3$ + Ours | $D^3$ | $D^3$ + Ours | $D^3$ | $D^3$ + Ours | $D^3$ |
| Resnet-18 | 2K | 0.6561 | **0.6663** | 0.7682 | 0.6750 | 0.7379 | 0.606 | 0.7509 | 0.6724 | 0.7380 | 0.5939 | **0.7488** | 0.6368 |
| | 4K | 0.6751 | **0.6812** | 0.7624 | 0.6853 | 0.7328 | 0.6494 | 0.7576 | 0.7028 | 0.7208 | 0.6163 | **0.7434** | 0.6635 |
| | 6K | 0.6780 | **0.6995** | 0.7886 | 0.6870 | 0.7493 | 0.6586 | 0.7768 | 0.7285 | 0.7349 | 0.6335 | **0.7624** | 0.6769 |
| | 8K | 0.6828 | **0.6964** | 0.7710 | 0.6741 | 0.7454 | 0.6418 | 0.7785 | 0.7186 | 0.7215 | 0.6033 | **0.7541** | 0.6595 |
| | 10K | 0.6830 | **0.6957** | 0.7807 | 0.6897 | 0.7483 | 0.6682 | 0.7781 | 0.7326 | 0.7300 | 0.6229 | **0.7593** | 0.6784 |
| ViT-T | 2K | **0.6759** | 0.6672 | 0.7550 | 0.6827 | 0.7585 | 0.6758 | 0.7473 | 0.6941 | 0.7327 | 0.6106 | **0.7484** | 0.6658 |
| | 4K | **0.6878** | 0.6871 | 0.7576 | 0.7000 | 0.7605 | 0.7071 | 0.7549 | 0.7217 | 0.7221 | 0.6144 | **0.7488** | 0.6858 |
| | 6K | **0.6898** | 0.6891 | 0.7663 | 0.6962 | 0.7666 | 0.7164 | 0.7629 | 0.7238 | 0.7303 | 0.6134 | **0.7565** | 0.6875 |
| | 8K | 0.6962 | **0.6974** | 0.7655 | 0.6894 | 0.7712 | 0.7253 | 0.7653 | 0.7253 | 0.7381 | 0.6344 | **0.7600** | 0.6936 |
| | 10K | **0.6986** | 0.6984 | 0.7828 | 0.6960 | 0.7777 | 0.7275 | 0.7786 | 0.7334 | 0.7330 | 0.6293 | **0.7680** | 0.6966 |

## F.2 Results

Table 9 and Table 8 evaluate the performance of ResNet-18 and ViT-T detection backbones trained on datasets of varying sizes and compositions across in-domain (same model and cross-model) and out-of-domain settings. While models trained with $D^3$ and GENERATE ANY SCENE occasionally underperform compared to those trained solely on $D^3$ in the in-domain same-model setting, they exhibit significant advantages in both in-domain cross-model and out-of-domain evaluations. These results demonstrate that incorporating our data (GENERATE ANY SCENE) into the training process

enhances the detector's robustness. By supplementing existing datasets with GENERATE ANY SCENE under the same training configurations and dataset sizes, detectors achieve stronger cross-model and cross-dataset capabilities, highlighting improved generalizability to diverse generative models and datasets.

Table 9: F1-Score Comparison of ResNet-18 and ViT-T Detectors Trained with $D^3$ and $D^3$+ GENERATE ANY SCENE Across Out-of-Domain Settings

| Detector | Data Scale (2N) | SDv1.5 | | VQDM | | Midjourney | | Average (Out-of-domain) | |
|---|---|---|---|---|---|---|---|---|---|
| | | $D^3$ + Ours | $D^3$ | $D^3$ + Ours | $D^3$ | $D^3$ + Ours | $D^3$ | $D^3$ + Ours | $D^3$ |
| Resnet-18 | 2K | 0.6515 | 0.6591 | 0.5629 | 0.5285 | 0.5803 | 0.5647 | **0.5982** | 0.5841 |
| | 4K | 0.6709 | 0.6817 | 0.5693 | 0.5428 | 0.6016 | 0.5941 | **0.6139** | 0.6062 |
| | 6K | 0.6750 | 0.6963 | 0.5724 | 0.5327 | 0.6084 | 0.6072 | **0.6186** | 0.6121 |
| | 8K | 0.6792 | 0.6965 | 0.5716 | 0.5282 | 0.6097 | 0.5873 | **0.6202** | 0.6040 |
| | 10K | 0.6814 | 0.6955 | 0.5812 | 0.5454 | 0.6109 | 0.6040 | **0.6245** | 0.6150 |
| ViT-T | 2K | 0.6755 | 0.6685 | 0.5443 | 0.4966 | 0.6207 | 0.6066 | **0.6135** | 0.5906 |
| | 4K | 0.6845 | 0.6865 | 0.5591 | 0.4971 | 0.6416 | 0.6149 | **0.6284** | 0.5995 |
| | 6K | 0.6900 | 0.6890 | 0.5580 | 0.4948 | 0.6455 | 0.6259 | **0.6313** | 0.6032 |
| | 8K | 0.6940 | 0.6969 | 0.5553 | 0.4962 | 0.6495 | 0.6387 | **0.6329** | 0.6106 |
| | 10K | 0.6961 | 0.6988 | 0.5499 | 0.4975 | 0.6447 | 0.6358 | **0.6302** | 0.6107 |

## G    Limitation

**Programmatically generated prompts can be unrealistic and biased.**    Programmatically generated prompts can be unrealistic and biased. Although our system is capable of producing a wide range of rare compositional scenes and corresponding prompts, some of these outputs may violate rules or conventions, going beyond what is even considered imaginable or plausible. We also implement a pipeline to filter the commonsense of the generated prompts using the *Vera score* (a large language model-based commonsense metric) and *Perplexity*, but we make this pipeline **optional**.

**Linguistic diversity of programmatic prompts is limited.**    While GENERATE ANY SCENE excels at generating diverse and compositional scene graphs and prompts, its ability to produce varied language expressions is somewhat constrained. The programmatic approach to generating content ensures diversity in terms of the elements of the scene, but it is limited when it comes to linguistic diversity and the richness of expression. To address this, we introduce a pipeline that leverages large language models (LLMs) to paraphrase prompts, enhancing linguistic variety. However, this addition introduces new challenges. LLMs are prone to biases and hallucinations, which can affect the quality and reliability of the output. Furthermore, the use of LLMs risks distorting the integrity of the original scene graph structure, compromising the coherence and accuracy of the generated content. So we make this LLM paraphrase pipeline **optional** for our paper.

**Toward curriculum-aware GRPO training.**    Our proposed GENERATE ANY SCENE framework plays a central role in GRPO training by providing structured scene graphs that serve as the foundation for a semantically grounded and controllable reward function. This design enables effective optimization by aligning generation objectives with fine-grained visual semantics. Beyond this, we also observe that GENERATE ANY SCENE also offers broader potential: the scene graphs it produces vary in complexity, such as in the number of objects, attributes, relationships and graph degree. These variations naturally correspond to different levels of generation difficulty and reward variance. This property suggests an opportunity for curriculum-based training, where the model could be progressively exposed to increasingly complex scene graphs. Such a strategy may improve training stability and efficiency, especially in the early stages of learning. We identify this as a promising direction for future work, further leveraging the controllability of GENERATE ANY SCENE to guide structured policy learning.

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

[88] Kling AI. Kling ai text-to-video. https://klingai.com/text-to-video/new, 2025. Accessed May 23, 2025.

[89] Team Wan, Ang Wang, Baole Ai, Bin Wen, Chaojie Mao, Chen-Wei Xie, Di Chen, Feiwu Yu, Haiming Zhao, Jianxiao Yang, Jianyuan Zeng, Jiayu Wang, Jingfeng Zhang, Jingren Zhou, Jinkai Wang, Jixuan Chen, Kai Zhu, Kang Zhao, Keyu Yan, Lianghua Huang, Mengyang Feng, Ningyi Zhang, Pandeng Li, Pingyu Wu, Ruihang Chu, Ruili Feng, Shiwei Zhang, Siyang Sun, Tao Fang, Tianxing Wang, Tianyi Gui, Tingyu Weng, Tong Shen, Wei Lin, Wei Wang, Wei Wang, Wenmeng Zhou, Wente Wang, Wenting Shen, Wenyuan Yu, Xianzhong Shi, Xiaoming Huang, Xin Xu, Yan Kou, Yangyu Lv, Yifei Li, Yijing Liu, Yiming Wang, Yingya Zhang, Yitong Huang, Yong Li, You Wu, Yu Liu, Yulin Pan, Yun Zheng, Yuntao Hong, Yupeng Shi, Yutong Feng, Zeyinzi Jiang, Zhen Han, Zhi-Fan Wu, and Ziyu Liu. Wan: Open and advanced large-scale video generative models. *arXiv preprint arXiv:2503.20314*, 2025.

[90] Meshy AI. Meshy ai – text-to-3d, image-to-3d, and text-to-texture 3d model generator. https://www.meshy.ai, 2025. Accessed May 23, 2025.

[91] Jiacheng Liu, Wenya Wang, Dianzhuo Wang, Noah A. Smith, Yejin Choi, and Hannaneh Hajishirzi. Vera: A general-purpose plausibility estimation model for commonsense statements, 2023.

[92] Xiaoshi Wu, Yiming Hao, Keqiang Sun, Yixiong Chen, Feng Zhu, Rui Zhao, and Hongsheng Li. Human preference score v2: A solid benchmark for evaluating human preferences of text-to-image synthesis. *arXiv preprint arXiv:2306.09341*, 2023.

[93] Giuseppe Vecchio and Valentin Deschaintre. Matsynth: A modern pbr materials dataset. In *Proceedings of the IEEE/CVF Conference on Computer Vision and Pattern Recognition*, pages 22109–22118, 2024.

[94] Sean Bell, Paul Upchurch, Noah Snavely, and Kavita Bala. Material recognition in the wild with the materials in context database. In *Proceedings of the IEEE conference on computer vision and pattern recognition*, pages 3479–3487, 2015.

[95] Jia Xue, Hang Zhang, Kristin Dana, and Ko Nishino. Differential angular imaging for material recognition. In *Proceedings of the IEEE Conference on Computer Vision and Pattern Recognition*, pages 764–773, 2017.

[96] Mircea Cimpoi, Subhransu Maji, Iasonas Kokkinos, Sammy Mohamed, and Andrea Vedaldi. Describing textures in the wild. In *Proceedings of the IEEE conference on computer vision and pattern recognition*, pages 3606–3613, 2014.

[97] Zhe Xu, Dacheng Tao, Ya Zhang, Junjie Wu, and Ah Chung Tsoi. Architectural style classification using multinomial latent logistic regression. In *Computer Vision–ECCV 2014: 13th European Conference, Zurich, Switzerland, September 6-12, 2014, Proceedings, Part I 13*, pages 600–615. Springer, 2014.

[98] Chitwan Saharia, William Chan, Saurabh Saxena, Lala Li, Jay Whang, Emily L Denton, Kamyar Ghasemipour, Raphael Gontijo Lopes, Burcu Karagol Ayan, Tim Salimans, et al. Photorealistic text-to-image diffusion models with deep language understanding. *Advances in neural information processing systems*, 35:36479–36494, 2022.

[99] Babak Saleh and Ahmed Elgammal. Large-scale classification of fine-art paintings: Learning the right metric on the right feature. *arXiv preprint arXiv:1505.00855*, 2015.

[100] Colby Crawford. 1000 cameras dataset. https://www.kaggle.com/datasets/crawford/1000-cameras-dataset, 2018. Accessed: 2024-11-09.

[101] Zijie J. Wang, Evan Montoya, David Munechika, Haoyang Yang, Benjamin Hoover, and Duen Horng Chau. DiffusionDB: A large-scale prompt gallery dataset for text-to-image generative models. *arXiv:2210.14896 [cs]*, 2022.

[102] Soravit Changpinyo, Piyush Sharma, Nan Ding, and Radu Soricut. Conceptual 12M: Pushing web-scale image-text pre-training to recognize long-tail visual concepts. In *CVPR*, 2021.

[103] Kaiming He, Xiangyu Zhang, Shaoqing Ren, and Jian Sun. Deep residual learning for image recognition. In *Proceedings of the IEEE conference on computer vision and pattern recognition*, pages 770–778, 2016.