# OpenReview forum: "Generate Any Scene: Synthetic Training and Evaluation Data for Generating Visual Content"
_NeurIPS.cc/2025/Datasets_and_Benchmarks_Track — Submitted to NeurIPS 2025 Datasets and Benchmarks Track_

### Official Review · Reviewer_6vf4 · 2025-07-01

**Rating:** 5
**Confidence:** 4

**Summary:**

This paper introduces "Generate Any Scene",  a pipeline to automatically generate (1) scene graphs composed of object nodes, object attributes and relational edges, (2) coherent captions from generated scene graphs using a proposed algorithm and (3) a set of question-answer pairs from generated scene graphs.

The proposed pipeline to generate compositional captions and corresponding QA pairs has the following applications:
- Self-improve Text-2-Image (T2I) and Text-2-Video (T2V) models by using self-generated images/videos with high VQA scores [37]
- Distill favorable properties of proprietary models like DaLL-E 3 to open-source model.
- Reinforcement learning with a synthetic reward function based on a light-weight open-sourced VLM, i.e. Qwen2.5-VL-3B
- Benchmark and analyze strengths/weaknesses of pretrained T2I and T2V models

**Additional Feedback:**

N/A

**Dataset Code Accessibility:**

Partly

**Dataset Code Comments:**

The reviewer has access to sampled scene graphs and captions for images, videos and 3D.
The reviewer couldn't find the QA pairs.

The reviewer has access to the github codebase. Dataset reproduction seems feasible.

**Ethical Considerations:**

No, there are no or only very minor ethics concerns

**Final Justification:**

The authors have fully addressed all of my concerns. I'm happy to increase the score and recommend acceptance.

**Limitations Weaknesses:**

1/ In Step1, may the authors detail further the engine used to generate scene graphs, especially the "satisfying constraints" (L114-115)? Does the authors take into account the physical feasibility of generated compositions?

2/ The algorithm to translate scene graph to caption is missing (Step 4). Is there a risk that generated captions are unrealistic which make it difficult for T2I and T2V models to interpret?

3/ In Step 5, it's unclear if manually-designed templates are used to generate question-answer for each node and edge?

4/ In Fig.4 (a), what is the "Best Open-Source Model"? Does distilling DaLL-E 3 to this best open-source model translate into the same degree of improvement?

**Strengths Contributions:**

Although missing a few technical details, the proposed pipeline to generate compositional captions and QA pairs seems correct. This paper provides extensive experiments to validate the effectiveness of such captions.

---

> ### Author Rebuttal · Authors · 2025-07-31
>
> We sincerely thank the reviewer for the detailed and constructive feedback, as well as for recognizing the contributions and extensive experiments of our work. We deeply appreciate the reviewer’s effort to carefully examine the technical details of our pipeline and data. Below, we provide detailed explanations and additional experimental evidence addressing each question:
>
> > Q1: May the authors detail further the engine used to generate scene graphs, especially the "satisfying constraints"? Does the authors take into account the physical feasibility of generated compositions?
>
> Our engine's controllable generation capability enables constraint satisfaction through two complementary dimensions: structural constraints and content constraints.
> - Structural Constraints: We achieve fine-grained control over scene graph topology through several mechanisms: (1) We can specify the number of nodes, edges, and overall graph complexity; (2) We can control the degree distribution of nodes and manage graph connectivity properties. (3) Seed graph preservation: our engine supports inputting a seed graph as a structural template. The generation process preserves this seed graph as a subgraph within the larger generated scene graph, ensuring that the final output contains the specified structural relationships.
>
> This structural controllability enables targeted generation for specific applications. For instance, in Application 2, we deliberately control the engine to generate scene structures containing multiple object compositions to test compositional understanding.
> - Content Constraints: As detailed in Appendix B, we constructed a hierarchical taxonomy of metadata organized in tree structures. This enables our engine to: (1) generate scenes focused on particular domains or themes; (2) select objects at different levels of semantic granularity.
>
> For example, in Appendix 3, we demonstrate controlled generation across different thematic domains to systematically evaluate model capabilities in various visual concepts.
>
> Regarding physical feasibility, while our engine primarily targets compositional coverage and diversity (rather than strict physical realism), our hierarchical taxonomy enables filtering for plausible combinations when desired. Additionally, after generation we can further apply commonsense‑oriented metrics such as VERA Score to filter out captions if required.
>
> >Q2.1: The algorithm to translate scene graph to caption is missing (Step 4)
>
> Our caption generation employs carefully designed templates that account for the diversity of elements and metadata types. The templates provide clear structural support for describing object-relationship-object and object-attribute compositions, utilizing determiners and ordinal numbers to distinguish multiple instances of the same object type.
>
> As we acknowledge in our limitations discussion, this approach might lack highly diverse linguistic expressions compared to LLM-based paraphrasing. However, using LLM generation would significantly increase computational costs and reduce efficiency. Our template-based approach represents a deliberate balance between efficiency and linguistic accuracy.
> Our additional validation study empirically demonstrates that the choice between templated and LLM-generated captions does not affect overall model performance evaluation, supporting the robustness of our approach.
>
> We sampled 100 scene graphs from our dataset and generated corresponding captions using GPT-4o paraphrasing. This 100-sample subset was selected while preserving the original 10K dataset’s complexity balance.
> Then we evaluated all models again using the LLM-paraphrased versions, computing VQA scores for them. The results are as follows:
>
> | Model                  | Original Score | Paraphrased Score | Difference | Original Rank | Paraphrased Rank |
> |------------------------|----------------|-------------------|------------|---------------|------------------|
> | DALLE-3                | 0.6871         | 0.7542            | +0.0671    | 1             | 1                |
> | FLUX.1-schnell         | 0.6132         | 0.6648            | +0.0516    | 2             | 2                |
> | PixArt-Σ               | 0.6109         | 0.6159            | +0.0050    | 3             | 3                |
> | PixArt-α               | 0.6049         | 0.6043            | -0.0006    | 4             | 4                |
> | Playground v2.5        | 0.5734         | 0.5075            | -0.0659    | 5             | 8                |
> | Stable Diffusion 3     | 0.5579         | 0.5140            | -0.0439    | 6             | 7                |
> | FLUX.1-dev             | 0.5561         | 0.5024            | -0.0537    | 7             | 9                |
> | DeepFloyd IF XL        | 0.5397         | 0.5606            | +0.0209    | 8             | 5                |
> | Wuerstchen v2          | 0.5352         | 0.5014            | -0.0338    | 9             | 10               |
> | SDXL                   | 0.5328         | 0.5322            | -0.0006    | 10            | 6                |
> | Stable Diffusion v2-1  | 0.5282         | 0.4961            | -0.0321    | 11            | 11               |
>
> The Pearson correlation coefficient between model rankings on templated versus paraphrased captions is **0.9232**, indicating a very strong positive correlation.
>
> >Q2.2: Is there a risk that generated captions are unrealistic which make it difficult for T2I and T2V models to interpret?
>
> The generation of "unrealistic" scenes is not a limitation but a meaningful feature addressing real-world applications. Modern generative models like DALL-E and Sora have gained recognition precisely for producing imaginative compositions such as "An astronaut riding a horse on the moon." Such unrealistic compositions are essential for comprehensive model evaluation and improvement. They test models' compositional understanding and creative capabilities beyond memorization of common patterns. Rather than artifacts to eliminate, these challenging scenarios are critical for assessing and enhancing models' fundamental ability to synthesize novel visual concepts, which defines one of the core values of creative generative AI systems.
>
> >Q3: It's unclear if manually-designed templates are used to generate question-answer for each node and edge?
>
> Yes, we use manually-designed templates to generate question-answer pairs for each node and edge in the scene graph.
> Our template-based approach systematically converts each structural component of the scene graph into corresponding questions. For object nodes, we generate existence queries; for attribute nodes, we create property verification questions; and for relation edges, we formulate relational queries between connected objects. We traverse all nodes and edges in the scene graph, ensuring comprehensive coverage of all structural elements.
>
> >Q4: what is the "Best Open-Source Model"? Does distilling DaLL-E 3 to this best open-source model translate into the same degree of improvement?
>
> The “Best Open-Source Model” in Fig. 4(a) refers to Flux.1-schnell. This model is a distilled, fixed-weights version and is therefore not suitable for direct fine-tuning. For our targeted distillation experiments from DaLL-E 3, we instead used the closely related Flux.1-dev variant.
>
> Following the same targeted fine-tuning settings described in the paper, we distilled DaLL-E 3’s capabilities using its generated datasets for hard concepts and multi-object scenes, and then evaluated on corresponding test sets. The results are summarized below:
> |   Test Set   | Original | Fine-tuned |
> |------------------------|----------|------------|
> | Hard Concept Test Set  | 0.303    | 0.361      |
> | Multi-object Test Set  | 0.271    | 0.325      |
>
> These results demonstrate consistent improvements across both challenging benchmarks, indicate that our distillation strategy generalizes well to different model variants.
>
> > Q5: The reviewer has access to sampled scene graphs and captions for images, videos and 3D. The reviewer couldn't find the QA pairs.
>
> We appreciate the reviewer’s careful examination of our dataset and code, and we would like to further clarify the point about QA pairs. Our open-sourced codebase supports the complete functionality to reproduce them on demand: (1) The JSON-format scene graphs can be converted to `nx.DiGraph` using  `gas.captions_generation.utils.convert_json_to_sg`. (2) QA pairs are then generated by traversing nodes and edges of this DiGraph through `gas.models.gen_model.text2image_metric. ProgrammaticDSGTIFAScore._get_dsg_questions`.
>
> This design allows flexible and reproducible QA generation. In Application 3, QA pairs are generated “just-in-time” during reward calculation rather than stored as static data. We plan to additionally release a set of pre-generated QA pairs in a future dataset update to further facilitate reproducibility.

---

> ### Author Response · Authors · 2025-08-08
> **Follow-up on Author–Reviewer Discussion**
>
> Dear Reviewer 6vf4,
>
> We truly appreciate the time and effort you have already dedicated to reviewing our paper and reading our rebuttal. We understand that the author-reviewer discussion phase is nearing its close, and if your schedule allows, we would be happy to further clarify any remaining points or provide additional details on our responses.
>
> Thank you again for your thoughtful feedback and for helping us improve the paper.
>
> Best regards,
>
> The authors of submissions807

---

### Official Review · Reviewer_BXvC · 2025-07-02

**Rating:** 4
**Confidence:** 3

**Summary:**

The core contribution is the use of "scene graph programming" to systematically and scalably generate a vast and diverse set of prompts, spanning from realistic to highly imaginative compositions. This approach directly addresses a key limitation of existing benchmarks, which primarily rely on real-world image-caption pairs and thus fail to adequately test the compositional and imaginative capabilities of modern generative models.

**Dataset Code Accessibility:**

Yes

**Ethical Considerations:**

No, there are no or only very minor ethics concerns

**Final Justification:**

Thanks for authos feedback, my concerns are partially solved. I remain my rate to borderline accpet.

**Limitations Weaknesses:**

1. "Scene Graph to Caption" may have a bottleneck issue.
The translation from a structured scene graph to a natural language caption (Section 3.1, Step 4) is a critical step, yet it appears to be highly templated. The authors should discuss this limitation more explicitly. While they acknowledge limited linguistic diversity in Section 8, they could strengthen the paper by running a small-scale study comparing model rankings on their programmatic prompts versus human-written or LLM-paraphrased prompts that describe the same scene graphs. This would help validate that the benchmark's findings are generalizable.

2. The paper relies exclusively on automated metrics (CLIP Score, VQA Score, TIFA, etc.) to make claims about model performance, faithfulness, and human preference. However, these metrics are known to have their own biases and limitations, a point the authors astutely make themselves in the supplementary material (Appendix A.2). Thus, I think a  human evaluation study would improve experiments.

**Strengths Contributions:**

1. The "scene graph programming" approach is a major strength. Unlike methods that rely on LLMs to generate prompts, which can be difficult to control and may introduce biases, this programmatic approach offers high degrees of control, scalability, and transparency.
2. The release of a 10-million-caption dataset is a valuable contribution to the community.

---

> ### Author Rebuttal · Authors · 2025-07-31
>
> We sincerely thank the reviewer for recognizing our "scene graph programming" approach as a major strength and its systematic advantages in addressing gaps in existing methods. We also thank for carefully reviewing our supplementary materials. Following the recommendation, we conducted the validation study:
>
> > Q1: running a small-scale study comparing model rankings on their programmatic prompts versus human-written or LLM-paraphrased prompts that describe the same scene graphs.
>
> We sampled 100 scene graphs from our dataset and generated corresponding captions using GPT-4o paraphrasing. Importantly, this 100-sample subset was selected while preserving the original 10K dataset’s complexity balance and distribution.
>
> The prompt we used is as follows:
> ```text
> You are given a scene graph in JSON format, where:
> - "nodes" contain objects and their attributes,
> - "edges" describe relationships between objects or link attributes to objects.
>
> Your task:
> 1. Understand the semantic meaning of each node and edge.
> 2. Convert the graph into a natural language caption that describes the entire scene.
> 3. Include all objects, attributes, and relations from the graph, and strictly follow the graph structure.
> 4. Do not introduce new objects or relationships not present in the graph.
>
> Input:
> {scene_graph}
> ```
> We then evaluated all models again using the LLM-paraphrased captions, and computing VQA scores for them:
>
>
> | Model                  | Original Score | Paraphrased Score | Difference | Original Rank | Paraphrased Rank |
> |------------------------|----------------|-------------------|------------|---------------|------------------|
> | DALLE-3                | 0.6871         | 0.7542            | +0.0671    | 1             | 1                |
> | FLUX.1-schnell         | 0.6132         | 0.6648            | +0.0516    | 2             | 2                |
> | PixArt-Σ               | 0.6109         | 0.6159            | +0.0050    | 3             | 3                |
> | PixArt-α               | 0.6049         | 0.6043            | -0.0006    | 4             | 4                |
> | Playground v2.5        | 0.5734         | 0.5075            | -0.0659    | 5             | 8                |
> | Stable Diffusion 3     | 0.5579         | 0.5140            | -0.0439    | 6             | 7                |
> | FLUX.1-dev             | 0.5561         | 0.5024            | -0.0537    | 7             | 9                |
> | DeepFloyd IF XL        | 0.5397         | 0.5606            | +0.0209    | 8             | 5                |
> | Wuerstchen v2          | 0.5352         | 0.5014            | -0.0338    | 9             | 10               |
> | SDXL                   | 0.5328         | 0.5322            | -0.0006    | 10            | 6                |
> | Stable Diffusion v2-1  | 0.5282         | 0.4961            | -0.0321    | 11            | 11               |
>
> The Pearson correlation coefficient between model rankings on templated versus paraphrased captions is **0.9232**, indicating a very strong positive correlation.
>
> This validation study demonstrates strong consistency between the two approaches. Importantly, the top-performing models (DALLE-3, FLUX.1-schnell, PixArt-Σ, PixArt-α) maintain their rankings across both evaluation conditions, while the relative ordering of models remains largely consistent.
> This high correlation validates that our programmatic approach produces rankings that are generalizable and not artifacts of the templated caption generation. The slight variations observed (e.g., some mid-tier models showing small rank changes) are within expected bounds and do not affect the overall conclusions about model capabilities.
>
> We will include this validation study in the revised manuscript to address the concerns about the generalizability of our templated approach.
>
> >Q2: a human evaluation study would improve evaluation experiments
>
> We appreciate the reviewer's thoughtful suggestion. We acknowledge that all automated metrics have inherent biases and limitations; while VQAScore has demonstrated improved alignment with human judgment compared to other automated metrics [1], we recognize the value of human evaluation validation.
>
>
> Given the high cost of comprehensive human evaluation, we strategically designed a focused study. We selected 6 models (DALL·E 3, FLUX.1‑schnell, PixArt‑$\Sigma$, Playground v2.5, Stable Diffusion 3 Medium, and Stable Diffusion v2‑1) with diverse performance characteristics and recruited 3 human evaluators. Each evaluator assessed 40 caption-images groups, with 10 overlapping groups across all evaluators to measure inter-annotator agreement. Evaluators ranked the generated images based on both relevance to the caption and overall visual quality.
>
> **Inter-Annotator Reliability**: The 3 evaluators showed strong agreement on the 10 shared samples, with a Spearman correlation coefficient of **0.962**, demonstrating consistent human judgment criteria.
>
> **Human-Metric Alignment**: The correlation between human rankings and our VQA Score rankings is **0.918**, indicating strong alignment between automated and human evaluation:
> | Model | VQA Ranking | Human Avg Ranking |
> |-------|-------------|-------------------|
> | Dalle-3 | 1 | 1.30 |
> | FLUX.1-schnell | 2 | 2.93 |
> | PixArt-$\Sigma$ | 3 | 3.65 |
> | Playground v2.5 | 4 | 3.08 |
> | SD3 Medium | 5 | 4.23 |
> | SD v2-1 | 6 | 5.83 |
>
> This study validates that our VQA Score-based rankings closely align with human preferences. The consistency between automated metrics and human judgment strengthens confidence in our benchmark's ability to assess model performance in a manner that reflects human perception.
>
> We will incorporate these human evaluation results into the revised manuscript to provide stronger validation of our automated metric approach.
>
> ***Reference***
>
> *[1] Lin, Zhiqiu, et al. "Evaluating text-to-visual generation with image-to-text generation." European Conference on Computer Vision. Cham: Springer Nature Switzerland, 2024.*

---

> ### Author Response · Authors · 2025-08-06
>
> Dear Reviewer BXvC,
>
> Thank you very much for your thoughtful and constructive feedback. We truly value the points you raised and have carefully addressed them in our rebuttal. We are very eager to hear your thoughts and would be glad to clarify anything further if needed. We greatly appreciate the opportunity to continue this discussion and improve our work based on your insights.
>
> Best regards,
>
> The authors of submissions807

---

> > ### Author Response · Authors · 2025-08-08
> > **Follow-up on Author–Reviewer Discussion**
> >
> > Dear Reviewer BXvC,
> >
> > We sincerely appreciate the time and effort you have dedicated to reviewing our paper. We understand that the Author–Reviewer Discussion phase is approaching its close, and your insights are highly valuable for the paper improvement and the AC’s final decision process. If your schedule allows, we would be happy to address any remaining concerns or provide further clarifications on the points in our rebuttal before the discussion period ends.
> >
> > Thank you again for your constructive feedback and for helping us strengthen the paper.
> >
> > Best regards,
> >
> > The authors of submissions807

---

### Official Review · Reviewer_nihG · 2025-07-02

**Rating:** 5
**Confidence:** 3

**Summary:**

Generate Any Scene presents a fully automated engine that enumerates all possible scene-graph topologies over a large metadata library and transforms each graph into fluent English captions and exhaustive QA pairs. Without any human labeling, it produces unbounded synthetic data to (1) self-improve text-to-image models via iterative VQA-guided fine-tuning, (2) distill compositional strengths from proprietary models into open-source ones, (3) define a scene-graph–based QA reward for RLHF that outperforms CLIP similarity, and (4) train robust AI-generated content detectors across diverse models and domains.

**Dataset Code Accessibility:**

Yes

**Dataset Code Comments:**

The authors have made the entire engine publicly available on GitHub, including scene-graph templates, metadata files (objects, attributes, relations, scene attributes), caption and QA generation scripts, and example configuration. They also host the full synthetic output — captions, QA pairs, and a representative image subset — on Hugging Face, complete with metadata and usage instructions.

**Ethical Comments:**

Generate Any Scene relies entirely on synthetic, metadata‐only scene graphs, captions, and QA pairs without using any real images of individuals or sensitive content, so there are no privacy, consent, or bias‐amplification concerns directly tied to personal data.

**Ethical Considerations:**

No, there are no or only very minor ethics concerns

**Final Justification:**

All of my major concerns have been fully addressed, and as a result, I am raising my score by one level. I expect the authors to incorporate all reviewers' feedback thoroughly in the camera-ready submission.

**Limitations Weaknesses:**

**1. Contextual Data Shortcomings.**
The core Generate Any Scene engine enumerates and instantiates scene graphs purely by combinatorial rules, with no grounding in real-world co-occurrence statistics or commonsense knowledge. As a result, it can produce implausible object pairings (e.g., “a person standing beside a wild bear in a domestic living room”) or semantically incoherent relations (e.g., “book on top of an airplane”). Because there are no frequency-based filters (e.g., from ConceptNet or Visual Genome) and no subject–predicate–object compatibility checks, many generated graphs describe scenes that defy basic logic. Moreover, without any post-generation plausibility verification, these unrealistic graphs remain in the dataset, diluting its overall quality and potentially teaching downstream models to expect events that never occur in real photographs.

**2. Methodological & Benchmark Gaps.**
In the RLHF experiments (Section 5), the authors compare only two reward signals—CLIP similarity vs. synthetic QA accuracy — without including standard VQA datasets (e.g., VQAv2, GQA) under the same training framework. This leaves it unclear whether the observed gains stem from the scene-graph–generated QA specifically or simply from using any QA-based reward. Likewise, the self-improvement fine-tuning studies pit SOS captions solely against CC3M real captions, omitting comparisons to other real-world caption sources (e.g., MS-COCO, LAION) that could contextualize the claimed ~4% improvement. Finally, the distillation experiments transfer DALL·E 3’s strengths via SOS pairs but do not benchmark against alternative data-mixing or augmentation strategies — such as combining web-crawled caption–image pairs — making it difficult to assess whether the proposed pipeline offers unique benefits beyond simpler approaches.

**Strengths Contributions:**

- The pipeline can generate virtually unlimited, richly annotated captions and QA pairs without any manual effort, covering rare and complex scene compositions.
- It applies this synthetic data across diverse downstream applications — iterative self-improvement, compositional distillation, fine-grained RLHF reward, and robust content detection — consistently boosting performance over real-data baselines and CLIP-based methods.

---

> ### Author Rebuttal · Authors · 2025-07-31
>
> We sincerely thank the reviewer for the comprehensive feedback and recognition of our pipeline's ability to generate "virtually unlimited, richly annotated captions and QA pairs" that "consistently boost performance over real-data baselines." We address your concerns below:
>
> > Q1: Concern on Data Realism
>
> We appreciate the reviewer's concern about plausibility.
>
> We believe generating “imaginary” or “ implausible” scenes is not a limitation but a meaningful feature that addresses the needs of real‑world generative applications. As demonstrated by the success of models like DALL-E and Sora, which gained attention for producing implausible images such as "An astronaut riding a horse on the moon," such creative compositions are widely valued in domains like art, game design, and film production. More importantly, diverse compositional combinations are essential for evaluating and enhancing models' generative and creative capabilities. These seemingly “implausible” images are not noise to be discarded but critical elements for comprehensive assessment. Our approach is specifically designed to meet this need for diverse captions and systematic visual representations, enabling coverage of both realistic and imaginative scenes.
>
> At the same time, our data generation engine is fully controllable and incorporates multiple filtering mechanisms. We systematically implement various filters including perplexity scores, VERA scores, complexity and other metrics that control generation based on linguistic coherence, commonsense, and scene complexity. Regarding frequency-based filtering specifically, we maintain comprehensive frequency statistics of all metadata objects across the LAION-5B dataset, enabling systematic metadata selection and caption filtering based on real-world occurrence patterns. Appendix A.3.1 of our appendix presents systematic comparative experiments across different filtering factors.
>
> Also, our metadata is organized into a systematic tree structure covering taxonomies at different granularities (Appendix B), providing fine-grained control over content topics. As demonstrated in Appendix Figure 14, we can effectively control and evaluate model performance across specific thematic domains.
>
> To address potential concerns about unrealistic content, we will release a controlled 10K caption version that emphasizes more realistic scenarios while maintaining our framework's systematic advantages.
>
> > Q2.1 : unclear whether the observed gains stem from the scene-graph–generated QA specifically or simply from using any QA-based reward.
>
> We thank the reviewer for pointing out this important comparison. To clarify this, we have added experiments incorporating manually annotated QA datasets - VQAv2, as additional reward signals under the same RLHF framework for direct comparison.
> We sampled 10K images from VQAv2, with corresponding QA pairs, matched them to COCO2017 captions, and applied same training frameworks to SimpleAR-0.5B-SFT with RL training. The results on GenAI Bench are shown in the table:
> | Method                             | Basic | Advanced | All |
> |------------------------------------|-------|----------|-----|
> | SimpleAR-0.5B-SFT                  | 0.74  | 0.60     | 0.66 |
> | SimpleAR-0.5B-RL (Clip)            | **0.75** | 0.60     | 0.67 |
> | SimpleAR-0.5B-RL (VQAv2)       | 0.73 | 0.59 | 0.66 |
> | **SimpleAR-0.5B-RL (Ours)**        | **0.75** | **0.61** | **0.68** |
>
> The results show that using VQAv2 captions and QA pairs as rewards yields even lower performance than CLIP‑based RL training. Furthermore, we observed minimal reward improvement from VQA signals throughout training. We attribute this to the fact that, although VQAv2 QA pairs are rich, the underlying image captions fail to cover enough visual elements, leading to a mismatch between QA pairs and captions that undermines RLHF reward alignment.
>
> This highlights the inherent difficulty and cost of constructing high-quality image-caption and QA annotations, whereas our method leverages scene-graph structures to systematically generate synthetic caption-QA pairs at minimal cost with unique advantages.
>
> >Q2.2: omitting comparisons to other real-world caption sources (e.g., MS-COCO, LAION) that could contextualize the claimed ~4% improvement
>
> We appreciate this suggestion and conducted more experiments comparing our approach to other real-world caption sources.
> We sampled 10K captions from MS-COCO-2017 and LAION-COCO for one-epoch LoRA fine-tuning under same experimental settings. The results on GAS test set are summarized below:
> | Baseline | GAS - High Scoring | GAS - Random Selection | COCO2017 | LIAON-5 | CC3M |
> |----------|---------------------|------------------------|----------|---------|------|
> | 0.508    | **0.530**               | 0.524                  | 0.508    | 0.510   | 0.508 |
>
> Fine‑tuning with MS‑COCO‑2017 and LAION‑COCO captions yields results similar to CC3M, with none surpassing the significant improvements achieved by our GAS captions.
> We think that although COCO2017 and LAION captions are generally high-quality and well‑aligned with images, they offer limited compositional diversity.
> These additional results confirm that the observed gains are not specific to CC3M but generalize across other widely used real-caption datasets. This further supports our claim that the compositional diversity of GAS synthetic captions drives the improvement.
>
> >Q2.3: Benchmark against alternative data-mixing or augmentation strategies — such as combining web-crawled caption–image pairs
>
> We conducted additional experiments to benchmark against alternative data sourcing strategies, specifically comparing our DALL-E 3 distillation approach with web-scraped real images.
> Using the Bing Image Search API, we retrieved images matching our multi‑object and hard‑concept captions and constructed two datasets of equivalent scale for comparison. We then applied the same fine‑tuning setup described in Application 2. The results are shown below:
> | Test Set               | Original | Fine-tuned with DALL·E 3 Distillation | Fine-tuned with Web-crawled Images |
> |------------------------|----------|--------------------------|------------------------------------|
> | Hard Concept Test Set  | 0.303    | **0.361**                  | 0.258                              |
> | Multi-object Test Set  | 0.271    | **0.325**                    | 0.264                              |
>
> The results show that web-scraped images not only failed to improve performance but actually degraded model capabilities.
>
> Upon examination of the retrieved images, we identified several critical issues. The web-crawled images contained significant noise, including watermarks, overlaid text, and irrelevant visual element. Our hard concept and multi-object captions feature high compositional complexity and novel object combinations that rarely exist in real-world photographs. The retrieved images showed poor relevance to our systematically designed compositional scenarios, as real-world images cannot adequately represent the diverse and controlled compositional variations we programmatically generate. Thus, training on such misaligned data appears to introduce incorrect visual-textual associations, leading to performance degradation rather than improvement.

---

> > ### Comment · Reviewer_nihG · 2025-08-07
> >
> > Thank you to the authors for their diligent work in preparing the rebuttal. All of my major concerns have been fully addressed, and as a result, I am raising my score by one level. If this paper is accepted, I expect the authors to incorporate all reviewers' feedback thoroughly in the camera-ready submission.

---

> > > ### Author Response · Authors · 2025-08-08
> > >
> > > Thank you for your positive update! We appreciate your constructive feedback and will ensure that all reviewers’ comments are carefully addressed in the next revision.

---

> > > ### Author Response · Authors · 2025-08-09
> > >
> > > Dear Reviewer nihG,
> > >
> > > As we near the end of the discussion phase, we wanted to sincerely thank you for your time, thoughtful feedback, and engagement. Your comments have been very helpful in improving our work, and we truly appreciate your support throughout the process.
> > >
> > > Best regards,
> > >
> > > The authors of submissions807

---

### Official Review · Reviewer_9mSj · 2025-07-02

**Rating:** 3
**Confidence:** 3

**Summary:**

This paper proposes GENERATE ANY SCENE, a data engine that can generate and translate infinite scene graphs into a caption for T2I and T2V, and also to a set of question-answering pairs for other use cases such as RLHF.  To generate any scene, the authors first collect 28787 objects, 1497 attributes and 10492 relations, 2193 scene attributes as the visual elements, and build scenes upon them. Then, the proposed data engine can translate the scene graphs into captions and QA pairs. Based on the proposed GENERATE ANY SCENE, the authors develop several use cases, including (1) self-improvement using generated data to improve T2Vision. For each caption, they use SD-1.5 to generate several images and then identify high-score ones. They use high-score generated images to finetune the model with LoRA. (2) distillation algorithm. They use GENERATE ANY SCENE to fine Dalle-3 is good at hard concepts and try to distill Dalle-3's compositionality to SD-1.5. (3) They use GRPO to improve SimpleAR-0.5B-SFT using synthesized captions and QA pairs from GENERATE ANY SCENE. The authors explore the GENERATE ANY SCENE in different scenarios, and show its effectiveness through multiple benchmarks on T2Vision generation.

**Dataset Code Accessibility:**

Yes

**Dataset Code Comments:**

The code and the data are provided.

**Ethical Considerations:**

No, there are no or only very minor ethics concerns

**Final Justification:**

I carefully read the rebuttal and the additional experiments provided by the authors. I agree that the proposed method is a controllable way to create synthetic data for training models. In the main paper, the authors show the results of their proposed self-improving method, and the backbone is SD-1.5. These results show that SD-1.5 fine-tuned using their proposed data engine is better than CC3M when the number of training samples is limited to 30K. However, the differences between CC3M baselines and their methods GAS are not that impressive (e.g., for GenAI-bench image, baseline=0.64x, GAS=0.65; for GenAI-bench video, baseline=0.65, GAS=0.65x). In the rebuttal, when the base model is changed to Flux, the differences between the CC3M baseline and GAS are also trivial (e.g., 0.538 v.s. 0.536, 0.737 v.s. 0.735). I am very happy to see this work explores synthetic data generation, but comparing with the complex synthetic data generation procedures, the gains are not satisfactory to me. Therefore, I decide to keep my original rating.

**Limitations Weaknesses:**

(1) In the iterative self-improving framework, the selection criteria is VQA Score and in Figure 2 the improvements are also shown using VQA Score. Do the authors have results on other evaluation metrics for these experiments to fully validate the framework?

(2) Still in the iterative self-improving framework, the authors fine-tune the model using LoRA. Have the authors tried to fully finetune the model and still see the results consistently improved?

(3) In each use case, the authors choose only one base model (e.g., SD-1.5). But this model is actually not SOTA and not strong enough comparing to some other open-sourced image generation models. Have the authors tried to use other base models and still see the improvements?

**Strengths Contributions:**

(1) The proposed use cases for GENERATE ANY SCENE are interesting. Using synthetic data for improving model performances is insightful.

(2) GENERATE ANY SCENE is controllable and can be reproduced using the provided taxonomy.

---

> ### Author Rebuttal · Authors · 2025-07-31
>
> We sincerely thank the reviewer for the constructive feedback and for recognizing the novelty of **Generate Any Scene**, particularly its controllable and scalable capability, and its broad applicability across diverse text-to-vision tasks.
>
> In the following, we address the reviewer’s concerns with additional experiments and analyses that further validate the effectiveness and generality of our approach.
>
> >Q1: Do the authors have results on other evaluation metrics for these experiments to fully validate the framework?
>
> Thanks for raising this important point. We initially adopted **VQA‑Score** as our primary selection and evaluation criterion due to its stronger correlation with human judgments of compositional image–text alignment than other metrics (CLIPScore, PickScore, ImageReward Score, TIFAScore, etc.), as demonstrated in prior work [1].
> To provide a more comprehensive comparison, we also evaluated other four different scoring metrics across the generated images: **TIFAScore, CLIPScore, ImageReward, and PickScore**. Key results are summarized in the table below:
>
> | Metric              | Baseline | Iter‑1 A | Iter‑1 B | Iter‑1 C | Iter‑2 A | Iter‑2 B | Iter‑2 C | Iter‑3 A | Iter‑3 B | Iter‑3 C |
> |---------------|---------:|---------:|---------:|---------:|---------:|---------:|---------:|---------:|---------:|---------:|
> | ImageReward Score | -0.797   |   -0.636   |   -0.650   |   -0.667   |   -0.604   |  -0.663  |   -0.622  |  -0.630   |   -0.623   |   -0.715   |
> | TIFAScore                |   0.213   |   0.222  |   0.222  |  0.221   |  0.233  |   0.223   |   0.236   |   0.240   |   0.233   |   0.241   |
> | CLIPScore                |   0.229  |   0.235   |  0.232  |   0.234   |  0.237   |   0.233   |   0.236   |  0.237   |  0.234   |   0.235  |
> | PickScore     |   0.190   |   0.193  |   0.192  |   0.192  |   0.192   |   0.191   |   0.192  |   0.191  |   0.191   |   0.191   |
>
> **A = GAS (Highest Scoring Selection);
> B = CC3M;
> C = GAS (Random Selection)*
>
> As shown in the table, fine-tuning with GAS captions consistently outperforms using CC3M data, with the most pronounced and sustained improvements observed on TIFA and ImageReward Scores. These gains demonstrate that self-improving with GAS captions provide stronger alignment between model outputs, textual semantics, and human preferences.
> Furthermore, the highest scoring selection strategy exhibits more stable improvements than random selection across most metrics, indicating that this strategy further enhances fine-tuning effectiveness. It is also worth noting that Pick Scores show only marginal improvements across all strategies, suggesting that these metrics are less sensitive to the fine-tuning variations considered here.
>
> Since VQAScore more faithfully reflects human alignment and semantic consistency than others, we still recommend prioritizing this metrics for model evaluation.
>
> >Q2: Have the authors tried to fully finetune the model and still see the results consistently improved?
>
> We thank the reviewer for this important question about full fine-tuning versus LoRA.
>
> We conducted full fine-tuning experiments across three iterations on Stable Diffusion 1.5. We compared three strategies: GAS captions with high‑score selection, GAS captions with random selection, and CC3M captions as the real‑data baseline.
> The results are summarized in the following two tables:
>
>
> **Result on GAS Test Set:**
> |  | Baseline | CC3M (Full FT) | GAS - High Scoring (Full FT) | GAS - Random Selection (Full FT) |
> |---|----------|----------------|-------------------------------|----------------------------------|
> | Iter‑1    | 0.508    | 0.496          | **0.510**    | 0.510                            |
> | Iter‑2    | -    | 0.518          | **0.534**   | 0.519                            |
> | Iter‑3    | -    | 0.519          | **0.540**  | 0.520                            |
>
> **Result on GenAI-Bench:**
>
> |  | Baseline | CC3M (Full FT) | GAS - High Scoring (Full FT) | GAS - Random Selection (Full FT) |
> |-----------|----------|----------------|-------------------------------|----------------------------------|
> | Iter‑1    | 0.617    | 0.589          | **0.620**    | 0.599                            |
> | Iter‑2    | -    | 0.619          | **0.626**   | 0.621                           |
> | Iter‑3    | -    | 0.622         | **0.634**  | 0.617                            |
>
> We found that using our GAS captions with high score selection not only improves performance consistently across iterations but also surpasses CC3M at every stage.
> The full fine-tuning results confirm that our captions and strategy's effectiveness is not dependent on the specific training approach (LoRA vs. full fine-tuning). The consistent improvement patterns across both evaluation benchmarks validate the robustness of our iterative self-improvement framework.
>
> >Q3: Have the authors tried to use other base models and still see the improvements?
>
> We acknowledge the reviewer's concern about our choice of base model and have conducted additional experiments using SOTA models to validate our approach's generalizability.
>
> We selected FLUX.1-dev, a current SOTA open-source model, as another base model.
>
> Firstly, We trained FLUX.1‑dev using the self‑improving framework from Application 1, following exactly the same experimental settings as used in our original experiments. Due to limited computational resources and time constraints, we performed one iteration of training. We then compared different data strategies, including GAS captions with high‑score selection, GAS captions with random selection, and CC3M captions, as summarized below:
>
> **Application 1 Results**:
> | Method | GAS-Test (VQA) | GenAI-Bench (VQA) |
> | ---------- | -------------- | ----------------- |
> |Baseline | 0.525 | 0.729 |
> | GAS - High Scoring | **0.538** | **0.737** |
> | GAS - Random Selection | 0.525 | 0.735 |
> | CC3M | 0.536 | 0.735 |
>
> We further applied our distillation framework (Application 2) to FLUX.1‑dev, using DALL‑E 3‑generated images of hard concepts and multi‑object captions to distill these capabilities into FLUX.1‑dev. The results are shown in the table below:
> |Test Set  | Original | Fine-tuned |
> |------------------------|----------|------------|
> | Hard concept test set  | 0.303    | **0.361**      |
> | Multi-object test set  | 0.271    | **0.325**      |
>
> The results demonstrate that our approach's effectiveness extends to state-of-the-art models. FLUX.1-dev shows consistent improvements across both applications: our top selection strategy achieves meaningful gains on both evaluation benchmarks, and the distillation approach yields substantial improvements on challenging compositional tasks.
>
> ***Reference***
>
> *[1] Lin, Zhiqiu, et al. "Evaluating text-to-visual generation with image-to-text generation." European Conference on Computer Vision. Cham: Springer Nature Switzerland, 2024.*

---

> > ### Comment · Reviewer_9mSj · 2025-08-06
> > **Thank you for providing additional results**
> >
> > I would like to thank the authors for providing additional results on different evaluation metrics and for the additional base model. Right now, I do not have more questions.

---

> > > ### Author Response · Authors · 2025-08-06
> > >
> > > We sincerely thank you for your time and positive feedback, and we are pleased that the additional results addressed your concerns.

---

> > > ### Author Response · Authors · 2025-08-09
> > >
> > > Dear Reviewer 9mSj,
> > >
> > > As the discussion phase is approaching its conclusion, we would like to once again express our sincere gratitude for the time and thought you have dedicated to reviewing our work and engaging in the discussion. We are pleased that the additional results and analyses have addressed your earlier concerns, and we would be grateful if you might consider reflecting this in your final evaluation. We also fully respect your independent judgment.
> > >
> > > Thank you again for your constructive feedback and for helping us improve the quality of our work.
> > >
> > > Best regards,
> > >
> > > The authors of submissions807

---

> ### Author Response · Authors · 2025-08-06
>
> Dear Reviewer 9mSj,
>
> Thank you very much for your thoughtful and detailed review of our submission. We truly appreciate the time and effort you have devoted to providing feedback. We have carefully addressed all your comments in our rebuttal, and also included additional experiments and analyses to further clarify the points you raised. We would be very grateful to know if our clarifications resolve any of the concerns you raised, or if there are additional points that would benefit from further discussion. We sincerely value your insights and would be happy to provide any further clarification that could assist in your evaluation.
>
> Thank you again for your constructive feedback and for helping improve our work.
>
> The authors of submissions807

---

### Comment · Area_Chair_wVdM · 2025-08-05
**reviewer-author discussion**

Dear reviewers,

The authors have responded to your reviews.  Please read the authors' rebuttal and other reviews, and actively participate in the author-reviewer discussion.

AC

---

### Note · Authors · 2025-08-14

We sincerely thank the AC and all reviewers for their careful assessment and constructive engagement. We appreciate the recognition that (1) GAS is an insightful, controllable, and scalable framework, grounded in our explicit taxonomy and scene graph programming, for synthesizing virtually unlimited, richly compositional T2Vision captions (**9mSj, nihG, BXvC**); and that (2) our synthetic captions deliver consistent gains across diverse applications compared with real-data baselines, supported by extensive experiments validating the effectiveness (**6vf4, nihG, 9mSj**).

We thank all reviewers for their feedback. During the rebuttal, we carefully addressed every concern by providing additional experiments, analyses and clarifications. After the discussion, we are pleased that Reviewer **nihG raised the score** and that Reviewer **9mSj reported no further concerns**. We also appreciate Reviewer **6vf4, BXvC thoughtful initial reviews**. We summarize the resulting improvements below.
1. **Plausibility and the value of imaginative scenes (nihG, 6vf4)**:
We clarified that GAS’s ability to generate both realistic and imaginative captions is crucial for unbiased, systematic model evaluation and improvement. We also emphasized GAS’s controllability by detailing its structural and content filtering mechanisms, which allow us to regulate the realism of captions and other properties.
2. **Broader baselines and comparative experiments (9mSj, nihG, 6vf4)**:
We expanded evaluation to additional metrics, training schemes, and SOTA models, confirming GAS’s consistent advantages. We compared against higher-quality and multi-source real-data training, showing the real-data limitations and highlighting the unique benefits of GAS synthetic captions.
3. **Human study and LLM‑paraphrase validation (BXvC)**:
We verified that model rankings are not template-dependent by paraphrasing captions for 100 GAS scene graphs with GPT-4o; rankings remained consistent. A focused human study further showed strong inter-annotator agreement and close alignment with VQA-based rankings.
4. **Technical detail clarifications (6vf4)**:
We provided precise clarifications on caption control and filtering mechanisms, scene-graph parsing, QA generation, and related terminology, ensuring full reproducibility and transparency of the pipeline.

We thank the reviewers again for their insightful feedback, and will reflect all resulting improvements in the revised submission.

---

### Decision · Program_Chairs · 2025-09-18

**Decision:**

Reject

**Comment:**

This paper proposes GENERATE ANY SCENE, a data engine that can generate and translate infinite scene graphs into a caption for T2I and T2V, and also into a set of question-answering pairs for other use cases such as RLHF. Without any human labeling, GENERATE ANY SCENE produces unbounded synthetic data to (1) self-improve T2I models via iterative VQA-guided fine-tuning, (2) distill compositional strengths from proprietary models into open-source ones, (3) define a scene-graph–based QA reward for RLHF that outperforms CLIP similarity, and (4) train robust AI-generated content detectors across diverse models and domains.  The concerns raised by the reviewers are insufficient evaluation metric for full validation, unclear self-improvement performance through iterations, weak baseline selection, unclear data realism, insufficient analysis of the obtained results, and undetailed description of the engine.  In the rebuttal, the authors addressed the raised concerns with providing additional evaluation results. Three of the four reviewers are mostly satisfied with the rebuttal, and two of them raised their ratings from BA to A. On the other hand, Reviewer 9mSj did not raise any more question in the author-reviewer discussions, but is not satisfied with unimpressive gain compared with the complex data generation procedures, keeping his/her rating BR unchanged.  Unlike methods relying on LLMs to generate prompts, the proposed method is a controllable way to create synthetic data for training models. Using synthetic data for iteratively self-improving the model brings insight to the community.  Although the performance improvements are not so significant, on balance, the contribution of this paper outweighs this concern. This paper should be accepted, accordingly.  All the discussion during the rebuttal and post-rebuttal discussion should be appropriately incorporated in the final version.

===== FINAL UPDATE FROM DB Track PCs ====

The final decision for this paper has been taken by the program chairs after consultation with the SACs. All Senior Area Chairs have ranked papers according to the feedback from the AC during the review process. We decided to leave the original meta-review to reflect the opinion of the AC in light of the initial discussions with reviewers and SAC.